# Unveiling Extraneous Sampling Bias with Data Missing-Not-At-Random

**Chunyuan Zheng**[1] **Haocheng Yang**[2] **Haoxuan Li**[1,*] **Mengyue Yang**[3,*]

[1]Peking University [2]National University of Singapore [3]University of Bristol

cyzheng@stu.pku.edu.cn

## Abstract

Selection bias poses a widely recognized challenge for unbiased evaluation and learning in many industrial scenarios. For example, in recommender systems, it arises from the users' selective interactions with items. Recently, doubly robust and its variants have been widely studied to achieve debiased learning of prediction models, however, all of them consider a simple exact matching scenario, i.e., the units (such as user-item pairs in a recommender system) are the same between the training and test sets. In practice, there may be limited or even no overlap in units between the training and test. In this paper, we consider a more practical scenario: the joint distribution of the feature and rating is the same in the training and test sets. Theoretical analysis shows that the previous DR estimator is biased even if the imputed errors and learned propensities are correct in this scenario. In addition, we propose a novel super-population doubly robust estimator (SuperDR), which can achieve a more accurate estimation and desirable generalization error bound compared to the existing DR estimators, and extend the joint learning algorithm for training the prediction and imputation models. We conduct extensive experiments on three real-world datasets, including a large-scale industrial dataset, to show the effectiveness of our method. The code is available at https://github.com/ChunyuanZheng/neurips-25-SuperDR.

## 1 Introduction

Selection bias means the distribution of collected data differs from that in the target population. It is ubiquitous and occurs when data are missing-not-at-random (MNAR). For example, in the recommender system (RS), due to the subjective preferences of users and the data collection process itself, selection bias always exists in the collected data [1, 2]. Thus, selection bias poses a widely-recognized challenge [3, 4, 5, 6]. Ignoring selection bias makes machine learning methods difficult to achieve unbiased predictions and reducing its reliability [7, 8].

Many methods have been proposed to address selection bias. The error imputation-based (EIB) method [9, 10] utilizes an imputation model to impute the missing relevance. The inverse propensity score (IPS) method uses inverse propensity to reweight the observed events to achieve unbiasedness [11, 12, 13]. The doubly robust (DR) method combines the error imputation model and the propensity model [14, 15, 16, 17, 18, 19], which is unbiased if either the imputed errors or the learned propensities are accurate, and is also proved to have lower variance compared to IPS [20].

Although previous DR-based methods have demonstrated promising performance in debiasing tasks, all of them consider the exact matching scenario, i.e., the units are the same between training and test set. In RS, it means the users and items are the same between training and test set, as illustrated in the left part in Figure 1. However, there may be limited or even no overlap in units between training and

---

*Corresponding authors

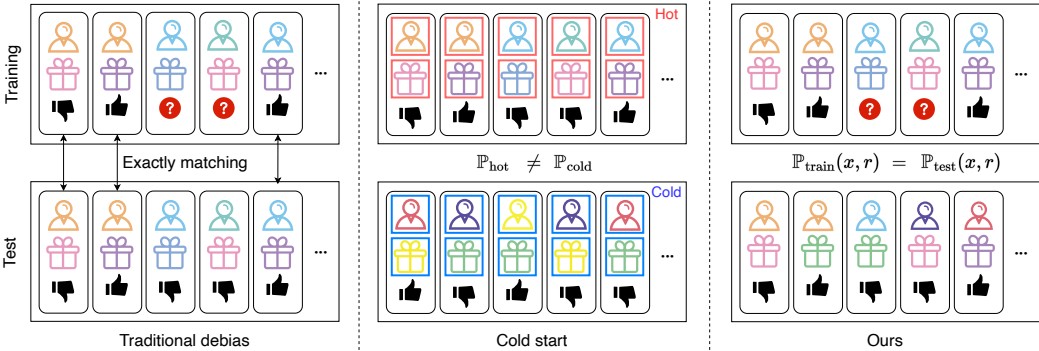

Figure 1: (1) Traditional debiasing methods consider an exact matching scenario, where the users and items are the same between training and test set, the non-random missingness (missing two negative ratings) indicates selection bias; (2) The cold-start problem refers to the training data containing only hot users/items, while the test data has only cold users/items, which differs from the debiasing scenario intrinsically; (3) This paper extends the exact matching scenario, considering a more general scenario that the joint distribution of the feature and rating $\mathbb{P}(x, r)$ in training and test set is the same.

test sets. For example, in many industrial scenarios, the user-item pairs in offline training data cannot be exactly the same as those in online test data [21, 22]. Therefore, instead of exact matching, a more practical scenario is that the joint distribution of the feature and outcome $\mathbb{P}(x, r)$ is the same in training and test set, as illustrated in the right part in Figure 1. Note that this scenario is intrinsically different with the cold-start problem, where the training data distribution $\mathbb{P}_{\text{hot}}$ differs from the test data distribution $\mathbb{P}_{\text{cold}}$, as shown in the middle part in Figure 1. In addition, selection bias is rarely considered in the cold-start problem.

To this end, in this paper, we first derive the bias of the DR estimator in super-population scenario, which contains an additional covariance term besides the term that measures the accuracy of imputed errors and learned propensities. Surprisingly, the DR estimator is biased even if the imputed errors and learned propensities are correct. Then we provide unbiased conditions under the super-population scenario, and propose the SuperDR estimator with the corrected imputation model, which can effectively control the additional covariance term with many desirable theoretical properties such as bias and variance reduction, leading to a more accurate estimation under the super-population scenario. In addition, we extend the previous joint learning algorithm based on the proposed corrected imputation loss and further derive the generalization error bound for the proposed SuperDR, and show that the proposed learning approach can effectively control it. Extensive experiments are conducted on three real-world datasets to show the effectiveness of our SuperDR method.

Our main contributions can be summarized as follows:

- To the best of our knowledge, this is the first paper that extends the exact matching scenario to the super-population scenario. In this scenario, we derive the bias of the DR estimator and show DR estimator is biased even if the imputed errors and learned propensities are correct.
- We propose the SuperDR method based on the corrected imputation model, which can effectively control the additional covariance term. In addition, the corrected imputation model not only benefits unbiased estimation but also reduces the generalization bound to enhance prediction performance.
- We conduct extensive experiments on three real-world datasets, including a large industrial dataset, to demonstrate the effectiveness of our proposed method.

## 2 Related Work

There are various biases in the collected data [23, 24], which have been of increasing concern in recent years [25, 26, 27, 28, 29, 30, 31]. Selection bias is one of the most common biases and a lot of research has been done aiming to eliminate this kind of bias [32, 33, 34, 35, 36]. Based on the causal inference techniques [37, 38, 39, 40, 41, 42, 43], the error imputation based method (EIB) [44, 45] first imputes pseudo-labels for missing events from the observed events, and then leverages these pseudo-labels to train the prediction model [46]. The propensity-based approaches weight the inverse

propensity score (IPS) on the observed data to eliminate bias [11, 12, 13]. However, IPS will suffer from a large variance when the extreme values exist in the estimated propensities [47].

The doubly robust (DR) method improves the weakness of EIB and IPS methods and becomes the mainstream model due to the weaker unbiased conditions (unbiased when either imputed errors or learned propensities are correct) and smaller variance than the IPS method [48, 49, 50, 51, 52]. In particular, the DR estimator is unbiased when either learned propensities or imputed errors are accurate. Many augmented DR methods are developed to further enhance the previous DR method performance by modifying the propensity model and imputation model or the form of the DR estimator, such as MRDR [33], BRD-DR [53], StableDR [54], TDR [55], DR-MSE [56], MR [57], CDR [58], AKBDR [59], DCE-TDR [60], and D-DR [61]. In addition, there are methods leveraging few unbiased ratings to mitigate hidden confounding and improve DR debiasing efficacy [5, 62, 63]. In this paper, we consider a more general super-population scenario and propose SuperDR with the corrected imputation model to achieve a more accurate estimation.

## 3 Preliminaries

We start with the classic debiasing scenario and take RS as an example. Note that the selection bias also exists in other scenarios such as pattern recognition and causal effects estimation, and our proposed method is also applicable for debiasing in these scenarios. Suppose the training user set $\mathcal{U}_{\text{train}} = \{u_1, u_2, \ldots, u_m\}$ contains $m$ users, the item set $\mathcal{I}_{\text{train}} = \{i_1, i_2, \ldots, i_n\}$ contains $n$ items. The purpose of RS is to train a prediction model to accurately predict the ratings of all user-item pairs, thus the target population is defined as all user-item pairs $\mathcal{D}_{\text{target}} = \mathcal{U}_{\text{train}} \times \mathcal{I}_{\text{train}}$. Let $\mathbf{R} \in \mathbb{R}^{m \times n}$ be the ground truth rating matrix of all user-item pairs in $\mathcal{D}_{\text{target}}$, where $r_{u,i}$ is the rating of user $u$ on item $i$. Let $x_{u,i}$ be the feature of user $u$ and item $i$, and $\hat{r}_{u,i} = f(x_{u,i}; \theta)$ is the predicted rating by a prediction model, $\theta$ is the corresponding parameter. Denote $\hat{\mathbf{R}} \in \mathbb{R}^{m \times n}$ as the matrix containing all the predicted ratings. Let $\mathbf{O} \in \{0, 1\}^{m \times n}$ be the binary observation indicator matrix for all user-item pairs, $o_{u,i} = 1$ indicates the rating of user $u$ on item $i$ is observed, otherwise missing $o_{u,i} = 0$. All previous methods implicitly assume $\mathcal{D}_{\text{target}} = \mathcal{D}_{\text{test}} = \mathcal{D}$ with fixed user-item pairs, thus the only randomness comes from the missing mechanism. If all the ratings are observed, the prediction model can be trained directly by minimizing the following ideal loss

$$\mathcal{L}_{\text{ideal}}(\theta) = \frac{1}{|\mathcal{D}|} \sum_{(u,i) \in \mathcal{D}} e_{u,i},$$

where $e_{u,i} = \mathcal{L}(\hat{r}_{u,i}, r_{u,i})$ is the loss between the predicted rating $\hat{r}_{u,i}$ and the true rating $r_{u,i}$ and $\mathcal{L}(\cdot, \cdot)$ is an arbitrary loss function. However, the ideal loss is not available in most cases because we can only observe part of the data with selection bias. Thus, for user-item pair with $o_{u,i} = 0$, the $r_{u,i}$ is missing. To tackle this issue, the DR estimator has been proposed:

$$\mathcal{E}_{\text{DR}}(\theta) = \frac{1}{|\mathcal{D}|} \sum_{(u,i) \in \mathcal{D}} \left[ \hat{e}_{u,i} + \frac{o_{u,i}(e_{u,i} - \hat{e}_{u,i})}{\hat{p}_{u,i}} \right].$$

where $\hat{p}_{u,i} = \pi(x_{u,i}; \psi)$ is the propensity model to estimate $p_{u,i} := \mathbb{P}(o_{u,i} = 1 \mid x_{u,i})$, and $\hat{e}_{u,i} = m(x_{u,i}; \phi)$ is the imputation model to impute the missing $e_{u,i}$.

## 4 Proposed Method

### 4.1 From Finite Population to Super-population

Before introducing our method, we first focus on the theoretical properties of the DR estimator and start from the bias form of DR estimator.

**Lemma 4.1** (Bias of DR Estimator [14]). *Given imputed errors $\hat{e}_{u,i}$ and learned propensities $\hat{p}_{u,i} > 0$, when considering only the randomness of missing indicators, the bias of DR estimator is*

$$\text{Bias}_{\mathbf{O}}[\mathcal{E}_{\text{DR}}(\theta)] = \frac{1}{|\mathcal{D}|} \sum_{(u,i) \in \mathcal{D}} \frac{\{\hat{p}_{u,i} - p_{u,i}\} \cdot \{e_{u,i} - \hat{e}_{u,i}\}}{\hat{p}_{u,i}}.$$

We find that either $\hat{e}_{u,i} = e_{u,i}$ or $\hat{p}_{u,i} = p_{u,i}$ is sufficient to eliminate bias, which inspires the double robustness condition for the DR method.

**Corollary 4.2** (Double Robustness [14])**.** *The DR estimator is unbiased when either imputed errors $\hat{e}_{u,i}$ or learned propensities $\hat{p}_{u,i}$ are accurate for all user-item pairs, i.e., either $\hat{e}_{u,i} = e_{u,i}$ or $\hat{p}_{u,i} = p_{u,i}$ for all $u$ and $i$.*

The above Lemma 4.1 shows the bias form of the DR estimator when users and items in the training set and the users and items in the test set are exactly the same. However, as we discussed earlier, this scenario is too simple in many real-world scenarios. Thus, we consider a more general scenario, also known as super-population, with $\mathcal{U} = \{u_1, u_2, ...\}, \mathcal{I} = \{i_1, i_2, ...\}$. $\mathcal{D}_{\text{target}} = \{u_1, u_2, \ldots, u_m\} \times \{i_1, i_2, \ldots, i_n\}$ and $\mathcal{D}_{\text{test}} = \{u_{j_1}, u_{j_2}, \ldots, u_{j_{m'}}\} \times \{i_{k_1}, i_{k_2}, \ldots, i_{k_{n'}}\}$ are sampled from the whole user set and item set, respectively. Without loss of generality, we consider the sampling strategy to be the same for both $\mathcal{D}_{\text{target}}$ and $\mathcal{D}_{\text{test}}$ datasets (otherwise, we can adjust the sampling strategy by reweighting). Therefore, instead of $\mathcal{D}_{\text{target}} = \mathcal{D}_{\text{test}} = \mathcal{D}$, we consider a more practical scenario $\mathbb{P}(\mathcal{D}_{\text{target}}) = \mathbb{P}(\mathcal{D}_{\text{test}}) = \mathbb{P}(\mathcal{D})$, that is, the joint distribution $\mathbb{P}(x, r)$ in the target and the test population are the same. We can regard the ground-truth ratings and covariate values as drawing $|\mathcal{D}_{\text{target}}|$ times from the $\mathbb{P}(x, r)$ in the super-population and are therefore they are stochastic. Furthermore, the randomness of ratings and covariates leads to the randomness of all other variables such as $e_{u,i}$ and $\hat{e}_{u,i}$. For unbiased prediction in this scenario, we need to estimate the expected ideal loss below:

$$\mathcal{L}^*_{\text{ideal}}(\theta) = \mathbb{E}[\mathcal{L}_{\text{ideal}}(\theta)] = \mathbb{E}[e_{u,i}],$$

where the expectation is taken on the super-population distribution $\mathbb{P}(x, r)$. Unless otherwise stated, all expectations are taken on the $\mathbb{P}(x, r)$ later. With the additional randomness caused by super-population, the theoretical results of the DR estimator change. The following theorem and corollary show the bias and the double robustness property under super-population for the DR estimator.

**Theorem 4.3** (Bias of DR Estimator under **Super-population**)**.** *Given error imputation model $\hat{e}_{u,i}$ and propensity model $\hat{p}_{u,i}$, then the bias of the DR estimator for estimating the expected ideal loss under super-population is*

$$\text{Bias}_{\mathcal{P}}[\mathcal{E}_{\text{DR}}(\theta)] = \underbrace{\text{Cov}\left(\frac{\hat{p}_{u,i} - o_{u,i}}{\hat{p}_{u,i}}, e_{u,i} - \hat{e}_{u,i}\right)}_{\textit{equals to 0 if independent}} + \underbrace{\mathbb{E}\left[\left\{1 - \mathbb{E}\left[\frac{o_{u,i}}{\hat{p}_{u,i}}\Big|x_{u,i}\right]\right\} \cdot \left\{\mathbb{E}[e_{u,i} \mid x_{u,i}] - \mathbb{E}[\hat{e}_{u,i} \mid x_{u,i}]\right\}\right]}_{\textit{equals to 0 either } \mathbb{E}[o_{u,i}/\hat{p}_{u,i} \mid x_{u,i}] = 1 \textit{ or } \mathbb{E}[\hat{e}_{u,i} - e_{u,i} \mid x_{u,i}] = 0}.$$

**Corollary 4.4** (Double Robustness under **Super-population**)**.** *Under super-population, the DR estimator is unbiased when both the following conditions hold:*

*(i) Either learned propensities satisfy $\mathbb{E}[o_{u,i}/\hat{p}_{u,i} \mid x_{u,i}] = 1$, or imputed errors have the same conditional expectation with true prediction errors $\mathbb{E}[\hat{e}_{u,i} \mid x_{u,i}] = \mathbb{E}[e_{u,i} \mid x_{u,i}]$;*

*(ii) The covariance term vanishes, that is, $\text{Cov}\left(\frac{\hat{p}_{u,i} - o_{u,i}}{\hat{p}_{u,i}}, e_{u,i} - \hat{e}_{u,i}\right) = 0$.*

**Remark:** Previous DR estimators are biased even if $\hat{e}_{u,i} = e_{u,i}$ or $\hat{p}_{u,i} = p_{u,i}$ for all $(u, i) \in \mathcal{D}_{\text{target}}$.

Compared with the existing theoretical results as in Lemma 4.1, it is obvious that condition *(i)* is necessary to achieve unbiasedness, which directly extends the conditions of accurate imputed errors and learned propensities in Lemma 4.1 to the expectation form. However, note that the condition *(ii)* that covariance vanishes is also needed for the unbiasedness under super-population scenario. Intuitively, if ignoring the randomness caused by sampling process, then $e_{u,i} - \hat{e}_{u,i}$ is a constant given $x_{u,i}$. By the double expectation formula, the covariance term vanishes automatically. The detailed proofs are in the Appendix A. Therefore, it is necessary to modify the previous DR learning approach to control the covariance while learning accurate propensity and imputation models under super-population scenario.

### 4.2 The SuperDR Estimator

It is important to note that the true covariance is unknown because we cannot access the true data distribution. However, we can use the empirical covariance over all user-item pairs as an approximation of the true covariance. We first give the definition of empirical covariance.

**Definition 4.5** (Empirical Covariance). The empirical expected conditional covariance between $(\hat{p}_{u,i} - o_{u,i})/\hat{p}_{u,i}$ and $e_{u,i} - \hat{e}_{u,i}$ is

$$\widehat{\text{Cov}}\left(\frac{\hat{p}_{u,i} - o_{u,i}}{\hat{p}_{u,i}}, e_{u,i} - \hat{e}_{u,i}\right) = \frac{1}{|\mathcal{D}|} \sum_{(u,i)\in\mathcal{D}} \frac{\hat{p}_{u,i} - o_{u,i}}{\hat{p}_{u,i}} \cdot (e_{u,i} - \hat{e}_{u,i}).$$

When the learned propensities or imputed errors are accurate, i.e., satisfying condition *(i)* in Corollary 4.4, the empirical covariance will converge to $\text{Cov}\left(\frac{\hat{p}_{u,i}-o_{u,i}}{\hat{p}_{u,i}}, e_{u,i} - \hat{e}_{u,i}\right)$ as $|\mathcal{D}| \to \infty$. A direct method to control the empirical covariance is to regard it as a regularization term. However, since the data are partially observed, we cannot obtain the value of the empirical covariance on all user-item pairs. In addition, the large penalty term may hurt the prediction performance. Interestingly, motivated by targeted maximum likelihood estimation [55, 64], we found that the empirical covariance can be controlled with a targeting correction step based on the DR estimator. Specifically, we designed imputation correction as follows:

$$\tilde{e}_{u,i} = \hat{e}_{u,i} + \epsilon(o_{u,i} - \hat{p}_{u,i}).$$

where $\hat{e}_{u,i} = m(x_{u,i}; \phi)$ is the imputed errors in previous DR estimators, $\hat{p}_{u,i} = \pi(x_{u,i}; \psi)$ is the learned propensity, and $\epsilon$ is a learnable parameter. We optimize $\phi$ and $\epsilon$ in $\tilde{e}_{u,i}$ by minimizing the loss based on imputation correction:

$$(\phi^*, \epsilon^*) = \arg\min_{\phi,\epsilon} \mathcal{L}_e^{Sup}(\phi, \epsilon) = \frac{1}{|\mathcal{D}|} \sum_{(u,i)\in\mathcal{D}} \frac{o_{u,i}(e_{u,i} - \tilde{e}_{u,i})^2}{\hat{p}_{u,i}}.$$

Specifically, the added correction term $\epsilon(o_{u,i} - \hat{p}_{u,i})$ has several desired properties. First, the correction term enlarges the hypothesis space of $\tilde{e}_{u,i}$ compared to $\hat{e}_{u,i}$, and does not bring extra concerns to the double robustness property due to it has zero mean under accurate $\hat{p}_{u,i}$. Second, the derivatives on the proposed loss with respect to $\epsilon$ are shown below:

$$\frac{\partial}{\partial\epsilon} \mathcal{L}_e^{Sup}(\phi, \epsilon) = \frac{2}{|\mathcal{D}|} \sum_{(u,i)\in\mathcal{O}} \frac{\hat{p}_{u,i} - o_{u,i}}{\hat{p}_{u,i}} \cdot (e_{u,i} - \tilde{e}_{u,i}).$$

It has the same form as the empirical covariance for user-item pairs with $o_{u,i} = 1$, which means that we can make the empirical covariance for observed user-item pairs to zero by minimizing the $\mathcal{L}_e^{Sup}$ directly. Note that adding the correction term on either $\hat{e}_{u,i}$ or $e_{u,i}$ will not affect the gradient above, thus we add such term on $\hat{e}_{u,i}$ for illustration. In the next step, we show that the unobserved empirical covariance can also be bounded by minimizing $\mathcal{L}_e^{Sup}$ using the concentration inequality. To proceed, we first define the empirical Rademacher complexity as follows.

**Definition 4.6** (Empirical Rademacher Complexity [65]). Let $\mathcal{F}$ be a family of prediction models mapping from $x \in \mathcal{X}$ to $[a, b]$, and $S = \{x_{u,i} \mid (u,i) \in \mathcal{D}\}$ a fixed sample of size $|\mathcal{D}|$ with elements in $\mathcal{X}$. Then, the empirical Rademacher complexity of $\mathcal{F}$ with respect to the sample $S$ is defined as:

$$\mathcal{R}(\mathcal{F}) = \mathbb{E}_{\boldsymbol{\sigma}\sim\{-1,+1\}^{|\mathcal{D}|}} \sup_{f_\theta \in \mathcal{F}} \left[ \frac{1}{|\mathcal{D}|} \sum_{(u,i)\in\mathcal{D}} \sigma_{u,i} e_{u,i} \right],$$

where $\boldsymbol{\sigma} = \{\sigma_{u,i} : (u,i) \in \mathcal{D}\}$, and $\sigma_{u,i}$ are independent uniform random variables taking values in $\{-1, +1\}$. The random variables $\sigma_{u,i}$ are called Rademacher variables.

Then we derive the controllability of empirical covariance for all user-item pairs in Theorem 4.7. Refer to Appendix A for the complete proof for this theorem.

**Theorem 4.7** (Controllability of Empirical Covariance). *The corrected imputation model trained by $\mathcal{L}_e^{Sup}$ is sufficient for controlling the empirical covariance.*

*(i) For user-item pairs with* **observed** *outcomes, the empirical covariance is 0. Formally, we have*

$$\frac{\partial}{\partial\epsilon} \mathcal{L}_e^{Sup}(\phi, \epsilon)\bigg|_{\epsilon=\epsilon^*} = 0, \ \text{which is equivalent to} \ \frac{1}{|\mathcal{D}|} \sum_{(u,i):\, o_{u,i}=1} \frac{\hat{p}_{u,i} - o_{u,i}}{\hat{p}_{u,i}} \cdot (e_{u,i} - \tilde{e}_{u,i}) = 0;$$

**Algorithm 1:** The Proposed Doubly Robust Joint Learning Algorithm under **Super-population**

---

**Input:** observed ratings $\mathbf{R}^o$ and a pre-trained propensity model $\pi(x_{u,i}; \psi)$.

**1 while** *stopping criteria is not satisfied* **do**

**2**     **for** *number of steps for training the corrected imputation model* **do**

**3**        Sample a batch of user-item pairs $\{(u_j, i_j)\}_{j=1}^J$ from $\mathcal{O}$;

**4**        Update $\phi$ by descending along the gradient $\nabla_\phi \mathcal{L}_e^{Sup}(\phi, \epsilon)$;

**5**        **Update $\epsilon$ by descending along the gradient $\nabla_\epsilon \mathcal{L}_e^{Sup}(\phi, \epsilon)$;**

**6**     **end**

**7**     **for** *number of steps for training the debiased prediction model* **do**

**8**        Sample a batch of user-item pairs $\{(u_k, i_k)\}_{k=1}^K$ from $\mathcal{D}$;

**9**        Update $\theta$ by descending along the gradient $\nabla_\theta \mathcal{L}_{\text{SuperDR}}(\theta; \phi, \psi)$;

**10**    **end**

**11 end**

---

*(ii) For user-item pairs with **missing** outcomes, suppose that $\hat{p}_{u,i} \geq K_\psi$ and $|e_{u,i} - \tilde{e}_{u,i}| \leq K_\phi$, then with probability at least $1 - \eta$, we have*

$$\frac{1}{|\mathcal{D}|} \sum_{(u,i): o_{u,i}=0} \frac{\hat{p}_{u,i} - o_{u,i}}{\hat{p}_{u,i}} \cdot (e_{u,i} - \tilde{e}_{u,i})$$

$$\leq \underbrace{\sqrt{\mathcal{L}_e^{Sup}(\phi, \epsilon)}}_{\text{proposed loss}} + K_\phi \underbrace{\sqrt{\frac{1}{|\mathcal{D}|} \sum_{u,i \in \mathcal{D}} \left| 1 - \mathbb{E}\left[ \frac{o_{u,i}}{\hat{p}_{u,i}} \Big| x_{u,i} \right] \right|}}_{\text{empirical bias from propensity model}} + \underbrace{\sqrt{K_\phi \left( 1 + \frac{1}{K_\psi} \right) \left( 2\mathcal{R}(\mathcal{F}) + (2K_\phi + 1) \sqrt{\frac{2 \log(4/\eta)}{|\mathcal{D}|}} \right)}}_{\text{tail bound controlled by empirical Rademacher complexity and sample size}},$$

*where the $K_\psi$, $K_\phi$, $\eta$ are constants.*

Note the proposed imputation correction has no harm property theoretically, as shown in Corollary 4.8.

**Corollary 4.8** (Relation to previous imputed errors). *The learned coefficient $\epsilon^*$ will converge to zero when the imputation model $\hat{e}_{u,i}$ has zero empirical covariance, making $\tilde{e}_{u,i}$ degenerates to $\hat{e}_{u,i}$.*

In addition, the proposed imputation correction can not only control the empirical covariance effectively but also be helpful for learning more accurate imputed errors.

**Corollary 4.9** (Bias reduction property). *The proposed corrected imputation loss leads to the smaller bias of imputed errors $\tilde{e}_{u,i}$, when $\hat{e}_{u,i}$ are inaccurate. Formally, we have*

$$\min_{\phi, \epsilon} \mathcal{L}_e^{Sup}(\phi, \epsilon) = \frac{1}{|\mathcal{D}|} \sum_{(u,i) \in \mathcal{D}} \frac{o_{u,i}(e_{u,i} - \tilde{e}_{u,i})^2}{\hat{p}_{u,i}} \leq \min_\phi \mathcal{L}_e(\phi) = \frac{1}{|\mathcal{D}|} \sum_{(u,i) \in \mathcal{D}} \frac{o_{u,i}(e_{u,i} - \hat{e}_{u,i})^2}{\hat{p}_{u,i}}.$$

Moreover, while reducing bias, the proposed method also reduces the variance compared to the previous imputed errors under a moderate condition, as shown below.

**Corollary 4.10** (Variance reduction property). *The proposed corrected imputation loss leads to the smaller variance of $\tilde{e}_{u,i}$ when the optimal $\epsilon^*$ lies in a certain range. Formally, we have*

$$\mathbb{V}(\tilde{e}_{u,i}) = \mathbb{V}(\hat{e}_{u,i} + \epsilon^* \cdot (o_{u,i} - \hat{p}_{u,i})) \leq \mathbb{V}(\hat{e}_{u,i}), \;\; \text{if } \epsilon^* \in \left[ 0, \; 2 \cdot \frac{\text{Cov}(\hat{e}_{u,i}, \hat{p}_{u,i} - o_{u,i})}{\mathbb{V}(\hat{p}_{u,i} - o_{u,i})} \right].$$

See Appendix A for the proof for the above three corollaries. Finally, the proposed SuperDR estimator is given below based on the corrected imputation:

$$\mathcal{E}_{\text{SuperDR}}(\theta) = \frac{1}{|\mathcal{D}|} \sum_{(u,i) \in \mathcal{D}} \left[ \tilde{e}_{u,i} + \frac{o_{u,i}(e_{u,i} - \tilde{e}_{u,i})}{\hat{p}_{u,i}} \right].$$

Table 1: Performance on AUC, NDCG@K and Recall@K on the **Coat**, **Yahoo! R3** and **KuaiRec** datasets. The best result is bolded and the best baseline result is underlined, where * means statistically significant results (p-value $\leq 0.05$) using the paired-t-test.

| Methods | Coat | | | Yahoo! R3 | | | KuaiRec | | |
|---|---|---|---|---|---|---|---|---|---|
| | AUC | NDCG@5 | Recall@5 | AUC | NDCG@5 | Recall@5 | AUC | NDCG@50 | Recall@50 |
| MLP | $0.729 \pm 0.003$ | $0.635 \pm 0.006$ | $0.614 \pm 0.007$ | $0.664 \pm 0.002$ | $0.645 \pm 0.002$ | $0.442 \pm 0.004$ | $0.808 \pm 0.005$ | $0.610 \pm 0.007$ | $0.645 \pm 0.010$ |
| DAMF | $0.729 \pm 0.005$ | $0.652 \pm 0.007$ | $0.628 \pm 0.008$ | $0.664 \pm 0.002$ | $0.642 \pm 0.001$ | $0.438 \pm 0.002$ | $0.811 \pm 0.003$ | $0.609 \pm 0.004$ | $0.643 \pm 0.005$ |
| CVIB | $0.729 \pm 0.004$ | $0.647 \pm 0.005$ | $0.623 \pm 0.009$ | $0.670 \pm 0.004$ | $0.656 \pm 0.003$ | $0.452 \pm 0.001$ | $0.816 \pm 0.007$ | $0.617 \pm 0.008$ | $0.653 \pm 0.009$ |
| IPS | $0.731 \pm 0.004$ | $0.642 \pm 0.004$ | $0.625 \pm 0.005$ | $0.667 \pm 0.003$ | $0.647 \pm 0.006$ | $0.445 \pm 0.007$ | $0.806 \pm 0.006$ | $0.606 \pm 0.006$ | $0.643 \pm 0.005$ |
| SNIPS | $0.732 \pm 0.004$ | $0.654 \pm 0.005$ | $0.629 \pm 0.005$ | $0.665 \pm 0.003$ | $0.644 \pm 0.004$ | $0.443 \pm 0.003$ | $0.811 \pm 0.004$ | $0.612 \pm 0.006$ | $0.649 \pm 0.006$ |
| ASIPS | $0.730 \pm 0.006$ | $0.643 \pm 0.006$ | $0.620 \pm 0.006$ | $0.668 \pm 0.002$ | $0.655 \pm 0.004$ | $0.452 \pm 0.005$ | $0.811 \pm 0.006$ | $0.614 \pm 0.006$ | $0.652 \pm 0.005$ |
| IPS-V2 | $0.736 \pm 0.004$ | $0.653 \pm 0.007$ | $0.628 \pm 0.009$ | $0.662 \pm 0.003$ | $0.651 \pm 0.001$ | $0.445 \pm 0.002$ | $0.813 \pm 0.006$ | $0.612 \pm 0.008$ | $0.655 \pm 0.006$ |
| DR | $0.733 \pm 0.003$ | $0.650 \pm 0.005$ | $0.625 \pm 0.007$ | $0.667 \pm 0.005$ | $0.655 \pm 0.004$ | $0.449 \pm 0.008$ | $0.818 \pm 0.003$ | $0.620 \pm 0.004$ | $0.655 \pm 0.007$ |
| MRDR | $0.739 \pm 0.005$ | $0.650 \pm 0.003$ | $0.622 \pm 0.007$ | $0.665 \pm 0.005$ | $0.652 \pm 0.005$ | $0.448 \pm 0.005$ | $0.814 \pm 0.006$ | $0.616 \pm 0.006$ | $0.652 \pm 0.003$ |
| DR-MSE | $0.738 \pm 0.005$ | $0.645 \pm 0.007$ | $0.627 \pm 0.006$ | $0.667 \pm 0.004$ | $0.650 \pm 0.004$ | $0.446 \pm 0.004$ | $0.814 \pm 0.006$ | $0.617 \pm 0.006$ | $0.654 \pm 0.007$ |
| DR-V2 | $0.747 \pm 0.004$ | $0.653 \pm 0.004$ | $0.625 \pm 0.006$ | $0.671 \pm 0.008$ | $0.660 \pm 0.005$ | $0.456 \pm 0.003$ | $0.821 \pm 0.010$ | $0.619 \pm 0.010$ | $0.661 \pm 0.008$ |
| SDR | $0.748 \pm 0.006$ | $0.650 \pm 0.005$ | $0.626 \pm 0.007$ | $0.666 \pm 0.005$ | $0.653 \pm 0.004$ | $0.451 \pm 0.004$ | $0.819 \pm 0.004$ | $0.618 \pm 0.005$ | $0.652 \pm 0.006$ |
| TDR | $0.744 \pm 0.004$ | $0.651 \pm 0.005$ | $\underline{0.631 \pm 0.005}$ | $0.664 \pm 0.004$ | $0.655 \pm 0.007$ | $0.453 \pm 0.003$ | $0.822 \pm 0.005$ | $0.621 \pm 0.009$ | $0.656 \pm 0.010$ |
| MR | $0.742 \pm 0.005$ | $0.653 \pm 0.006$ | $\underline{0.630 \pm 0.006}$ | $0.672 \pm 0.003$ | $0.657 \pm 0.003$ | $0.454 \pm 0.002$ | $0.823 \pm 0.003$ | $0.622 \pm 0.004$ | $0.655 \pm 0.005$ |
| AKBDR | $0.748 \pm 0.005$ | $\underline{0.656 \pm 0.007}$ | $0.630 \pm 0.007$ | $0.676 \pm 0.004$ | $\underline{0.662 \pm 0.004}$ | $\underline{0.461 \pm 0.004}$ | $0.824 \pm 0.004$ | $0.629 \pm 0.006$ | $0.667 \pm 0.006$ |
| DCE-TDR | $0.746 \pm 0.005$ | $\underline{0.654 \pm 0.005}$ | $0.629 \pm 0.006$ | $\underline{0.679 \pm 0.004}$ | $\underline{0.662 \pm 0.005}$ | $\underline{0.459 \pm 0.004}$ | $\underline{0.824 \pm 0.003}$ | $0.632 \pm 0.004$ | $0.671 \pm 0.006$ |
| D-DR | $\underline{0.750 \pm 0.004}$ | $0.654 \pm 0.004$ | $0.630 \pm 0.008$ | $0.678 \pm 0.004$ | $0.659 \pm 0.004$ | $0.456 \pm 0.003$ | $0.822 \pm 0.004$ | $\underline{0.630 \pm 0.005}$ | $\underline{0.672 \pm 0.005}$ |
| SuperDR | $\mathbf{0.757^* \pm 0.004}$ | $\mathbf{0.667^* \pm 0.005}$ | $\mathbf{0.637^* \pm 0.007}$ | $\mathbf{0.686^* \pm 0.003}$ | $\mathbf{0.667^* \pm 0.004}$ | $\mathbf{0.463 \pm 0.003}$ | $\mathbf{0.828^* \pm 0.004}$ | $\mathbf{0.640^* \pm 0.005}$ | $\mathbf{0.680^* \pm 0.005}$ |

## 4.3 The Extend Joint Learning Algorithm

We optimize the prediction model and the imputation model of the SuperDR method by a widely used joint learning framework [14], which alternatively optimizes two models to achieve unbiased learning. Specifically, we train the prediction model by minimizing SuperDR loss:

$$\mathcal{L}_{\text{SuperDR}}(\theta) = \frac{1}{|\mathcal{D}|} \sum_{(u,i) \in \mathcal{D}} \left[ \tilde{e}_{u,i} + \frac{o_{u,i}(e_{u,i} - \tilde{e}_{u,i})}{\hat{p}_{u,i}} \right].$$

We update the imputation model parameters and $\epsilon$ simultaneously by minimizing the $\mathcal{L}_e^{Sup}(\phi, \epsilon)$ in Section 4.2 and we train the propensity model by minimizing the following cross-entropy loss.

$$\mathcal{L}_p(\psi) = \frac{1}{|\mathcal{D}|} \sum_{(u,i) \in \mathcal{D}} \left[ -o_{u,i} \log(\hat{p}_{u,i}) - (1 - o_{u,i}) \log(1 - \hat{p}_{u,i}) \right].$$

The propensity model is pre-trained, and the parameters of the prediction and imputation model are updated alternatively via SGD. The joint learning process is summarized in Algorithm 1. Note that the complexity will not increase due to we only additionally update one single parameter $\epsilon$ compared to the traditional joint learning algorithm.

## 4.4 The Generalization Bound

Next, we analyze the generalization error bound of the DR methods using the models for estimating $e_{u,i}$ and $p_{u,i}$, and show that controlling empirical covariance leads to a tighter bound. Specifically, the generalization error theories for the previous DR estimators relied mainly on the boundedness of the loss to each user-item pair in the DR estimators from the binary indicator $o_{u,i}$, i.e., for the DR estimator, the bound for DR loss on $(u, i)$ is $(e_{u,i} - \hat{e}_{u,i})/\hat{p}_{u,i}$. However, these analyses no longer hold under super-population scenario. In the following theorem, we provide the generalization error bound of SuperDR, which includes four terms: the SuperDR loss, the empirical covariance, the bias of the SuperDR estimator, and the tail bound. Compared to previous DR methods, the proposed method can further control the covariance term, leading to a more desirable generalization bound thus improving debiasing performance. See Appendix A for the proof.

**Theorem 4.11** (Generalization Bound under **Super-population**). Suppose that $\hat{p}_{u,i} \geq K_\psi$ and $\min\{\tilde{e}_{u,i}, |e_{u,i} - \tilde{e}_{u,i}|\} \leq K_\phi$, then with probability at least $1 - \eta$, we have

$$\mathcal{L}_{ideal}(\theta) \leq \mathcal{L}_{\text{SuperDR}}(\theta) + \underbrace{\frac{1}{|\mathcal{D}|} \sum_{(u,i) \in \mathcal{D}} \left| 1 - \mathbb{E}\left[ \frac{o_{u,i}}{\hat{p}_{u,i}} \Big| x_{u,i} \right] \right| \cdot \left| \mathbb{E}[e_{u,i} \mid x_{u,i}] - \mathbb{E}[\tilde{e}_{u,i} \mid x_{u,i}] \right|}_{\text{vanilla DR only controls the empirical DR loss, and empirical risks of imputation and propensity models}}$$

$$+ \underbrace{\left| \frac{1}{|\mathcal{D}|} \sum_{(u,i) \in \mathcal{D}} \text{Cov}\left( \frac{o_{u,i} - \hat{p}_{u,i}}{\hat{p}_{u,i}}, e_{u,i} - \tilde{e}_{u,i} \right) \right|}_{\text{corrected loss further controls the independence}} + \underbrace{\left( 1 + \frac{1}{K_\psi} \right) \left( 2\mathcal{R}(\mathcal{F}) + K_\phi \sqrt{\frac{18 \log(4/\eta)}{|\mathcal{D}|}} \right)}_{\text{tail bound controlled by empirical Rademacher complexity and sample size}}$$

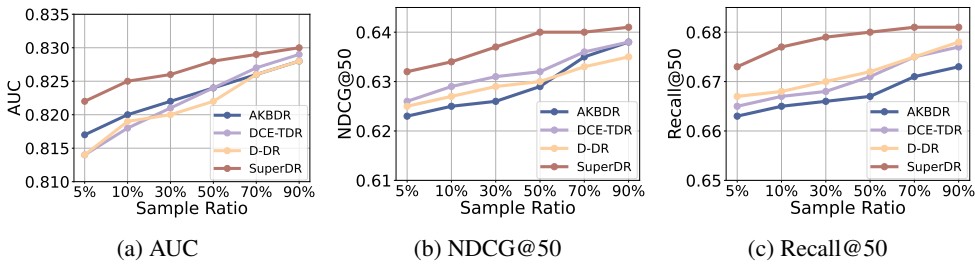

Figure 2: Effects of varying sample ratios $b\%$ on debiasing performance on the **KuaiRec** dataset.

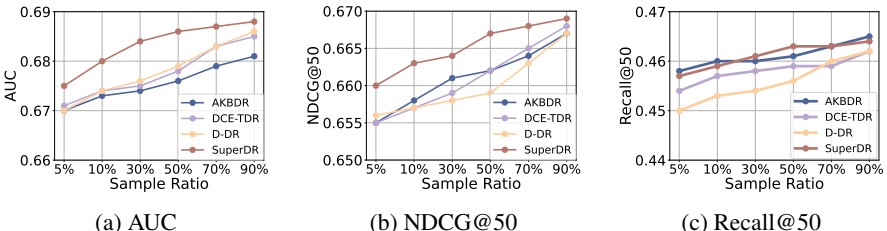

Figure 3: Effects of varying sample ratios $b\%$ on debiasing performance on the **Yahoo! R3** dataset.

## 5 Experiments

### 5.1 Experimental Setup

**Dataset Selection and Preprocessing.** To verify the effectiveness of the proposed method in the real-world dataset, the dataset that contains both biased and unbiased data is required. Following the previous studies [14, 15, 32, 66], the following three widely used real-world datasets are adopted to conduct our experiments: **Coat** contains ratings from 290 users to 300 items with 6,960 biased ratings and 4,640 unbiased ratings. **Yahoo! R3** contains ratings from 15,400 users to 1,000 items with 311,704 biased ratings and 54,000 unbiased ratings. We binarize the ratings to 0 for ratings less than three, otherwise to 1. We further use a fully exposed industrial dataset **KuaiRec** [67] with 4,676,570 video watching ratio records from 1,411 users to 3,327 videos. Following previous studies [59, 60], we biasedly select 201,171 samples according to the watch ratio as the training set and randomly select 117,113 samples as the unbiased test set. For this dataset, we binarize the records to 0 for records less than two, otherwise to 1

**Baselines.** In our experiments, as there are only very few features or no features for users and items in all three datasets, we first use the matrix factorization (MF) [3] method to generate the embedding for each user and item, and then fix such embedding as the user-item features. Then we take the MLP as the backbone model and compared the proposed method with the following debiasing baselines including **DAMF** [68], the information bottleneck based method: **CVIB** [69], the propensity based methods: **IPS** [13], **SNIPS** [70], **ASIPS** [35], and **IPS-V2** [71], and the DR-based methods: **DR** [14], **MRDR** [33], **DR-MSE** [56], **DR-V2** [71], **TDR** [55], **SDR** [54], **MR** [57], **AKBDR** [59], **DCE-TDR** [60], and **D-DR** [61].

**Experimental Protocols and Details.** The following three metrics are used to measure the debiasing performance: AUC, NDCG@K, and Recall@K, where we set K = 5 for **Coat** and **Yahoo! R3**, while set K = 50 for **KuaiRec**. All the experiments are implemented on PyTorch with the GeForce RTX 3090 as the computational resource. Adam is utilized as the optimizer in all experiments. To simulate the super-population scenario, we first randomly sample $b\%$ users and items (unless otherwise stated, $b$ is set to $50\%$ in our experiments) from the training set and then use the whole unbiased test set to evaluate the debiasing performance. Note that this intervention will not affect the data sparsity, it will only affect the number of observed users and items and will ensure $\mathbb{P}(\mathcal{D}_{\text{target}}) = \mathbb{P}(\mathcal{D}_{\text{test}})$ with limited overlapped users and items. In addition, the dimension of user and item embedding are fixed as 32. We tune learning rate in $\{0.001, 0.005, 0.01, 0.02, 0.05\}$ for parameters in prediction, imputation, and propensity model, and in $\{0.01, 0.05, 0.1, 0.15, 0.2\}$ for $\epsilon$, batch size in $\{128, 256, 512\}$ for **Coat** and $\{1024, 2048, 4096\}$ for **Yahoo! R3** and **KuaiRec**. The weight decay is tuned in $\{1e-6, 5e-6, \ldots, 5e-3, 1e-2\}$. In addition, we use the logistic regression model

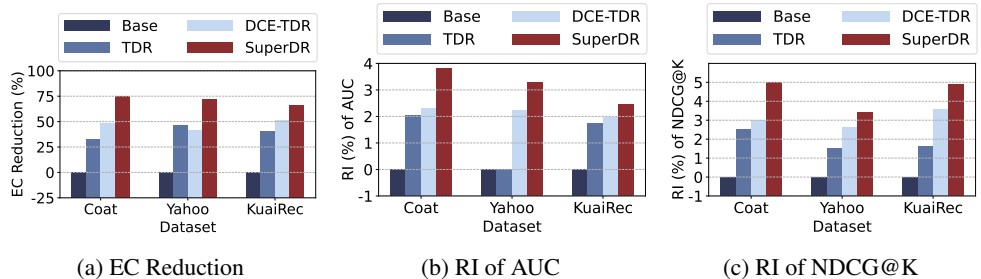

(a) EC Reduction          (b) RI of AUC          (c) RI of NDCG@K

Figure 4: Effects of empirical covariance (EC) reduction (%) on relative improvement (RI) (%).

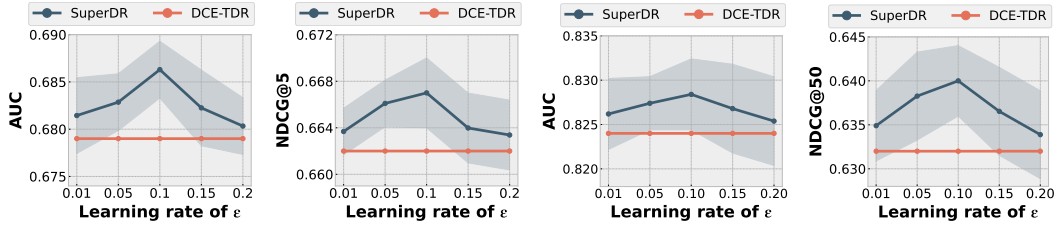

(a) AUC on Yahoo! R3    (b) NDCG@5 on Yahoo! R3    (c) AUC on KuaiRec    (d) NDCG@50 on KuaiRec

Figure 5: Effects of learning rate of the learnable imputation correction parameter $\epsilon$.

as the propensity model, which means that there is no unbiased data requirement. To prevent the propensity too small, we tune the propensity clip threshold in $[0.005, 0.05]$. For simplicity, we fix the step in inner loop for updating prediction and imputation models in Algorithm 1 as 1.

## 5.2 Performance Comparison

Table 1 summarizes the debiasing performance of various methods on three benchmark datasets **Coat**, **Yahoo! R3**, and **KuaiRec**, and we have the following findings. First, most debiased methods outperform the base model naive MLP, which shows the necessity for debiasing. Second, overall speaking, DR-based methods such as D-DR and DCE-TDR demonstrate the most competitive performance, indicating the superiority of DR methods over other baselines. Third, the proposed SuperDR method achieves the best performance in terms of all evaluation metrics. This indicates that the SuperDR method can effectively reduce the additional bias introduced by sampling through controlling empirical covariance, thus achieving an unbiased estimate of the ideal loss in scenarios where users and items in the training set are not exactly the same as those in the test set.

## 5.3 In-Depth Analysis

**Effects of Varying Bias Level.** Figures 2 investigates the impact of different levels of bias introduced by sampling on prediction performance on the **KuaiRec** dataset. We change the sample ratios to control the degree of overlap between users and items in the training and test sets. A higher sample ratio indicates a greater proportion of the same users and items in both sets, resulting in less bias introduced by sampling. When the sample ratio is large (e.g., 0.7 or 0.9), our method slightly outperforms recently proposed state-of-the-art methods such as DCE-TDR and D-DR. When the sample ratio is 0.05 or 0.1, there are few overlapping users and items between the training and test sets, resulting in significant bias introduced by sampling. The performance of previous methods noticeably declines, while the SuperDR method effectively addresses this bias, achieving significant performance improvements. We also conduct experiments on **Yahoo! R3** dataset. The experiment results are in Figure 3 with similar phenomenons.

**Effects of Empirical Covariance Control.** We explore the effects of Empirical Covariance (EC) Reduction on the prediction performance in Figure 4. We find that SuperDR achieves the most significant empirical covariance decreases and the most competitive performance in AUC and NDCG@K, which empirically demonstrates the effectiveness of the targeting correction step and the EC reduction benefit to the prediction performance. Note that DCE-TDR and TDR method obtains some performance improvement compared to base model naive MLP, this is because they

add $o_{u,i}(\frac{1}{\hat{p}_{u,i}} - 1)$ as the correction term to the imputed errors to control the covariance on observed samples. Unfortunately, DCE-TDR and TDR are unable to control the covariance on missing outcomes, resulting in sub-optimal performance.

### 5.4 Sensitivity Analysis

We conduct sensitivity analysis on the **Yahoo! R3** and **KuaiRec** datasets to explore the relationship between the learning rate of learnable parameter $\epsilon$ and the debiasing performance, with AUC and NDCG@K as the evaluation metrics, where K=5 on **Yahoo! R3** and K=50 on **KuaiRec**. As shown in Figure 5, the proposed SuperDR stably outperforms the DCE-TDR, the most competitive baseline on these two datasets, under varying learning rates of $\epsilon$, demonstrating that the enhanced imputation model with target learning mitigates the additional bias introduced by sampling and exhibits no-harm property. Meanwhile, under relatively moderate learning rates (0.05, 0.15), the SuperDR demonstrates competitive prediction performance, which further indicates the robustness.

## 6 Conclusion

In this paper, we extend the previous exact matching scenario, i.e., the units are the same between training and test set, and consider a more general scenario that the joint distribution of the feature and rating $\mathbb{P}(x, r)$ in the training and test set to be the same. Then we show the DR estimator is biased even if the imputed errors and learned propensities are correct in this scenario and provide the explicit bias form, which has two terms: the term that measures the accuracy of imputed errors and learned propensities and an additional covariance term. To achieve a more accurate estimation, we propose the SuperDR estimator with the corrected imputation model, which can effectively control the additional covariance term with many desirable theoretical properties such as bias and variance reduction. In addition, we extend the previous joint learning algorithm based on the proposed corrected imputation loss and further derive the generalization error bound for the proposed SuperDR, and show that the proposed learning approach can effectively control it. Extensive experiments are conducted on three real-world datasets to show the effectiveness of our SuperDR method. One of the potential limitations and research directions is how to develop a tighter bound for controlling the empirical covariance and to develop a more efficient algorithm for alternatively updating the prediction model, the imputation model, and the target learning parameter.

## Acknowledgments and Disclosure of Funding

This work is supported by the National Natural Science Foundation of China (623B2002).

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

# A  Proofs

**Lemma 4.1** (Bias of DR Estimator [14]). *Given imputed errors $\hat{e}_{u,i}$ and learned propensities $\hat{p}_{u,i} > 0$, when considering only the randomness of missing indicators, the bias of DR estimator is*

$$\text{Bias}_{\mathbf{O}}[\mathcal{E}_{\text{DR}}(\theta)] = \frac{1}{|\mathcal{D}|} \sum_{(u,i)\in\mathcal{D}} \frac{\{\hat{p}_{u,i} - p_{u,i}\} \cdot \{e_{u,i} - \hat{e}_{u,i}\}}{\hat{p}_{u,i}}.$$

*Proof of Lemma 4.1.* The proof can be found in Lemma 3.1 of [14]. However, one should note that, as stated in the proof, "the prediction and imputed errors are treated as constants when taking the expectation, since $o_{u,i}$ does not result from any prediction or imputation models [13]". The DR estimator in [14] is given as

$$\mathcal{E}_{\text{DR}}(\theta) = \frac{1}{|\mathcal{D}|} \sum_{(u,i)\in\mathcal{D}} \left[ \hat{e}_{u,i} + \frac{o_{u,i}(e_{u,i} - \hat{e}_{u,i})}{\hat{p}_{u,i}} \right].$$

By considering only the randomness on $o_{u,i}$, we have

$$\mathbb{E}_{\mathbf{O}}[\mathcal{E}_{\text{DR}}(\theta)] = \mathbb{E}_{\mathbf{O}}\left[ \frac{1}{|\mathcal{D}|} \sum_{(u,i)\in\mathcal{D}} \left[ \hat{e}_{u,i} + \frac{o_{u,i}(e_{u,i} - \hat{e}_{u,i})}{\hat{p}_{u,i}} \right] \right]$$

$$= \frac{1}{|\mathcal{D}|} \sum_{(u,i)\in\mathcal{D}} \left[ \hat{e}_{u,i} + \frac{p_{u,i}(e_{u,i} - \hat{e}_{u,i})}{\hat{p}_{u,i}} \right].$$

By definition, the bias of the DR estimator is

$$\text{Bias}_{\mathbf{O}}[\mathcal{E}_{\text{DR}}(\theta)] = \mathcal{E}_{ideal}(\theta) - \mathbb{E}_{\mathbf{O}}[\mathcal{E}_{\text{DR}}(\theta)]$$

$$= \frac{1}{|\mathcal{D}|} \sum_{(u,i)\in\mathcal{D}} e_{u,i} - \frac{1}{|\mathcal{D}|} \sum_{(u,i)\in\mathcal{D}} \left[ \hat{e}_{u,i} + \frac{p_{u,i}(e_{u,i} - \hat{e}_{u,i})}{\hat{p}_{u,i}} \right]$$

$$= \frac{1}{|\mathcal{D}|} \sum_{(u,i)\in\mathcal{D}} \frac{\{\hat{p}_{u,i} - p_{u,i}\} \cdot \{e_{u,i} - \hat{e}_{u,i}\}}{\hat{p}_{u,i}},$$

which yields the stated results. $\square$

**Corollary 4.2** (Double Robustness [14]). *The DR estimator is unbiased when either imputed errors $\hat{e}_{u,i}$ or learned propensities $\hat{p}_{u,i}$ are accurate for all user-item pairs, i.e., either $\hat{e}_{u,i} = e_{u,i}$ or $\hat{p}_{u,i} = p_{u,i}$ for all $u$ and $i$.*

*Proof of Corollary 4.2.* The proof can be found at Corollary 3.1 in Appendix of [14]. However, one should note that, as stated in the proof, "the prediction and imputed errors are treated as constants when taking the expectation, since $o_{u,i}$ does not result from any prediction or imputation models [13]".

Let $\delta_{u,i} = e_{u,i} - \hat{e}_{u,i}$ and $\Delta_{u,i} = \frac{\hat{p}_{u,i} - p_{u,i}}{\hat{p}_{u,i}}$. On the hand, when imputed errors are accurate, we have $\delta_{u,i} = 0$ for $(u,i) \in \mathcal{D}$. In such case, we can compute the bias of the DR estimator by

$$\text{Bias}_{\mathbf{O}}[\mathcal{E}_{\text{DR}}(\theta)] = \frac{1}{|\mathcal{D}|} \sum_{u,i\in\mathcal{D}} \Delta_{u,i}\delta_{u,i} = \frac{1}{|\mathcal{D}|} \sum_{u,i\in\mathcal{D}} \Delta_{u,i} \cdot 0 = 0.$$

On the other hand, when the learned propensities are accurate, we have $\Delta_{u,i} = 0$ for $(u,i) \in \mathcal{D}$. In this case, we can compute the bias of the DR estimator by

$$\text{Bias}\left(\mathcal{E}_{\text{DR}}\right) = \frac{1}{|\mathcal{D}|} \sum_{u,i\in\mathcal{D}} \Delta_{u,i}\delta_{u,i} = \frac{1}{|\mathcal{D}|} \sum_{u,i\in\mathcal{D}} 0 \cdot \delta_{u,i} = 0.$$

In both cases, the bias of the DR estimator is zero, which means that the expectation of the DR estimator over all the possible instances of $o_{u,i}$ is exactly the same as the prediction inaccuracy. This completes the proof. $\square$

**Theorem 4.3** (Bias of DR Estimator under **Super-population**). *Given error imputation model $\hat{e}_{u,i}$ and propensity model $\hat{p}_{u,i}$, then the bias of the DR estimator for estimating the expected ideal loss under super-population is*

$$\text{Bias}_{\mathcal{P}}[\mathcal{E}_{\text{DR}}(\theta)] = \underbrace{\text{Cov}\left(\frac{\hat{p}_{u,i} - o_{u,i}}{\hat{p}_{u,i}}, e_{u,i} - \hat{e}_{u,i}\right)}_{\text{equals to 0 if independent}} + \underbrace{\mathbb{E}\left[\left\{1 - \mathbb{E}\left[\frac{o_{u,i}}{\hat{p}_{u,i}}\Big|x_{u,i}\right]\right\} \cdot \left\{\mathbb{E}[e_{u,i} \mid x_{u,i}] - \mathbb{E}[\hat{e}_{u,i} \mid x_{u,i}]\right\}\right]}_{\text{equals to 0 either } \mathbb{E}[o_{u,i}/\hat{p}_{u,i} \mid x_{u,i}]=1 \text{ or } \mathbb{E}[\hat{e}_{u,i} - e_{u,i} \mid x_{u,i}]=0}.$$

*Proof of Theorem 1.* Instead of considering only the randomness of the rating missing indicator, in the following, we treat all variables, including imputed errors and learned propensities, as random variables. Formally, we have

$$\text{Bias}[\mathcal{E}_{\text{DR}}(\theta)] = \mathbb{E}[\mathcal{L}_{\text{ideal}}(\theta)] - \mathbb{E}[\mathcal{E}_{\text{DR}}(\theta)] = \mathbb{E}[e_{u,i}] - \mathbb{E}\left[e_{u,i} + \frac{\{o_{u,i} - \hat{p}_{u,i}\} \cdot \{e_{u,i} - \hat{e}_{u,i}\}}{\hat{p}_{u,i}}\right]$$

$$= \mathbb{E}\left[\mathbb{E}\left[\left\{\frac{\hat{p}_{u,i} - o_{u,i}}{\hat{p}_{u,i}}\right\}\{e_{u,i} - \hat{e}_{u,i}\} \mid x_{u,i}\right]\right] \qquad \text{(by the double expectation formula)}$$

$$= \mathbb{E}\left[\mathbb{E}\left[\left\{\frac{\hat{p}_{u,i} - o_{u,i}}{\hat{p}_{u,i}} - \mathbb{E}\left[\frac{\hat{p}_{u,i} - o_{u,i}}{\hat{p}_{u,i}}\right] + \mathbb{E}\left[\frac{\hat{p}_{u,i} - o_{u,i}}{\hat{p}_{u,i}}\right]\right\}\{(e_{u,i} - \hat{e}_{u,i}) - \mathbb{E}[e_{u,i} - \hat{e}_{u,i}] + \mathbb{E}[e_{u,i} - \hat{e}_{u,i}]\} \mid x_{u,i}\right]\right]$$

$$= \mathbb{E}\left[\mathbb{E}\left[\left\{\frac{\hat{p}_{u,i} - o_{u,i}}{\hat{p}_{u,i}} - \mathbb{E}\left[\frac{\hat{p}_{u,i} - o_{u,i}}{\hat{p}_{u,i}}\right]\right\}\{(e_{u,i} - \hat{e}_{u,i}) - \mathbb{E}[e_{u,i} - \hat{e}_{u,i}]\} \mid x_{u,i}\right]\right]$$

$$+ \mathbb{E}\left[\left\{1 - \mathbb{E}\left[\frac{o_{u,i}}{\hat{p}_{u,i}}\Big|x_{u,i}\right]\right\} \cdot \{\mathbb{E}[e_{u,i} \mid x_{u,i}] - \mathbb{E}[\hat{e}_{u,i} \mid x_{u,i}]\}\right]$$

$$= \text{Cov}\left(\frac{\hat{p}_{u,i} - o_{u,i}}{\hat{p}_{u,i}}, e_{u,i} - \hat{e}_{u,i}\right) + \mathbb{E}\left[\left\{1 - \mathbb{E}\left[\frac{o_{u,i}}{\hat{p}_{u,i}}\Big|x_{u,i}\right]\right\} \cdot \{\mathbb{E}[e_{u,i} \mid x_{u,i}] - \mathbb{E}[\hat{e}_{u,i} \mid x_{u,i}]\}\right],$$

which yields the stated results. $\qquad\square$

**Corollary 4.4** (Double Robustness under **Super-population**). *Under super-population, the DR estimator is unbiased when both the following conditions hold:*

*(i) Either learned propensities satisfy $\mathbb{E}[o_{u,i}/\hat{p}_{u,i} \mid x_{u,i}] = 1$, or imputed errors have the same conditional expectation with true prediction errors $\mathbb{E}[\hat{e}_{u,i} \mid x_{u,i}] = \mathbb{E}[e_{u,i} \mid x_{u,i}]$;*

*(ii) The covariance term vanishes, that is, $\text{Cov}\left(\frac{\hat{p}_{u,i} - o_{u,i}}{\hat{p}_{u,i}}, e_{u,i} - \hat{e}_{u,i}\right) = 0$.*

*Proof of Corollary 4.4.* First, when condition (ii) holds, i.e.,

$$\text{Cov}((\hat{p}_{u,i} - o_{u,i})/\hat{p}_{u,i}, e_{u,i} - \hat{e}_{u,i}) = 0,$$

it follows from the results in Theorem 1 that

$$\text{Bias}[\mathcal{E}_{\text{DR}}(\theta)] = \mathbb{E}\left[\left\{1 - \mathbb{E}\left[\frac{o_{u,i}}{\hat{p}_{u,i}}\Big|x_{u,i}\right]\right\} \cdot \{\mathbb{E}[e_{u,i} \mid x_{u,i}] - \mathbb{E}[\hat{e}_{u,i} \mid x_{u,i}]\}\right]$$

On the hand, when the learned propensities satisfy $\mathbb{E}[o_{u,i}/\hat{p}_{u,i} \mid x_{u,i}] = 1$. In such case, we can compute the bias of the DR estimator by

$$\text{Bias}[\mathcal{E}_{\text{DR}}(\theta)] = \mathbb{E}\left[0 \cdot \{\mathbb{E}[e_{u,i} \mid x_{u,i}] - \mathbb{E}[\hat{e}_{u,i} \mid x_{u,i}]\}\right] = 0.$$

On the other hand, when imputed errors have the same conditional expectation with true prediction errors, we have $\mathbb{E}[\hat{e}_{u,i} \mid x_{u,i}] = \mathbb{E}[e_{u,i} \mid x_{u,i}]$. In this case, we can compute the bias of the DR estimator by

$$\text{Bias}[\mathcal{E}_{\text{DR}}(\theta)] = \mathbb{E}\left[\left\{1 - \mathbb{E}\left[\frac{o_{u,i}}{\hat{p}_{u,i}}\Big|x_{u,i}\right]\right\} \cdot 0\right] = 0.$$

In both cases, the bias of the DR estimator is zero, which completes the proof. $\qquad\square$

**Definition 4.5** (Empirical Covariance). The empirical expected conditional covariance between $(\hat{p}_{u,i} - o_{u,i})/\hat{p}_{u,i}$ and $e_{u,i} - \hat{e}_{u,i}$ is

$$\widehat{\text{Cov}}\left(\frac{\hat{p}_{u,i} - o_{u,i}}{\hat{p}_{u,i}}, e_{u,i} - \hat{e}_{u,i}\right) = \frac{1}{|\mathcal{D}|}\sum_{(u,i)\in\mathcal{D}}\frac{\hat{p}_{u,i} - o_{u,i}}{\hat{p}_{u,i}} \cdot (e_{u,i} - \hat{e}_{u,i}).$$

**Definition 4.6** (Empirical Rademacher Complexity [65]). Let $\mathcal{F}$ be a family of prediction models mapping from $x \in \mathcal{X}$ to $[a, b]$, and $S = \{x_{u,i} \mid (u, i) \in \mathcal{D}\}$ a fixed sample of size $|\mathcal{D}|$ with elements in $\mathcal{X}$. Then, the empirical Rademacher complexity of $\mathcal{F}$ with respect to the sample $S$ is defined as:

$$\mathcal{R}(\mathcal{F}) = \mathbb{E}_{\boldsymbol{\sigma} \sim \{-1,+1\}^{|\mathcal{D}|}} \sup_{f_\theta \in \mathcal{F}} \left[ \frac{1}{|\mathcal{D}|} \sum_{(u,i) \in \mathcal{D}} \sigma_{u,i} e_{u,i} \right],$$

where $\boldsymbol{\sigma} = \{\sigma_{u,i} : (u, i) \in \mathcal{D}\}$, and $\sigma_{u,i}$ are independent uniform random variables taking values in $\{-1, +1\}$. The random variables $\sigma_{u,i}$ are called Rademacher variables.

**Lemma A.1** (Rademacher Comparison Lemma [65]). *Let $\mathcal{F}$ be a family of real-valued functions on $z \in \mathcal{Z}$ to $[a, b]$, and $S = \{x_{u,i} \mid (u, i) \in \mathcal{D}\}$ a fixed sample of size $|\mathcal{D}|$ with elements in $\mathcal{X}$. Then*

$$\mathbb{E}_{S \sim \mathbb{P}^{|\mathcal{D}|}} \left[ \sup_{f \in \mathcal{F}} \left( \mathbb{E}_{z \sim \mathbb{P}}[f(z)] - \frac{1}{|\mathcal{D}|} \sum_{(u,i) \in \mathcal{D}} f(z_{u,i}) \right) \right] \le 2 \mathbb{E}_{S \sim \mathbb{P}^{|\mathcal{D}|}} \mathbb{E}_{\boldsymbol{\sigma} \sim \{-1,+1\}^{|\mathcal{D}|}} \sup_{f \in \mathcal{F}} \left[ \frac{1}{|\mathcal{D}|} \sum_{(u,i) \in \mathcal{D}} \sigma_{u,i} f(z_{u,i}) \right],$$

*where $\boldsymbol{\sigma} = \{\sigma_{u,i} : (u, i) \in \mathcal{D}\}$, and $\sigma_{u,i}$ are independent uniform random variables taking values in $\{-1, +1\}$. The random variables $\sigma_{u,i}$ are called Rademacher variables.*

*Proof of Lemma A.1.* The proof can be found in Lemma 26.2 of [65]. □

**Lemma A.2** (McDiarmid's Inequality [65]). *Let $V$ be some set and let $f : V^m \to \mathbb{R}$ be a function of $m$ variables such that for some $c > 0$, for all $i \in [m]$ and for all $x_1, \ldots, x_m, x_i' \in V$ we have*

$$|f(x_1, \ldots, x_m) - f(x_1, \ldots, x_{i-1}, x_i', x_{i+1}, \ldots, x_m)| \le c$$

*Let $X_1, \ldots, X_m$ be $m$ independent random variables taking values in $V$. Then, with probability of at least $1 - \delta$ we have*

$$|f(X_1, \ldots, X_m) - \mathbb{E}[f(X_1, \ldots, X_m)]| \le c \sqrt{\log\left(\frac{2}{\delta}\right) m/2}$$

*Proof of Lemma A.2.* The proof can be found in Lemma 26.4 of [65]. □

**Lemma A.3** (Rademacher Calculus [65]). *For any $A \subset \mathbb{R}^m$, scalar $c \in \mathbb{R}$, and vector $\mathbf{a}_0 \in \mathbb{R}^m$, we have*

$$R(\{c\mathbf{a} + \mathbf{a}_0 : \mathbf{a} \in A\}) \le |c| R(A).$$

*Proof of Lemma A.3.* The proof can be found in Lemma 26.6 of [65]. □

**Theorem 4.7** (Controllability of Empirical Covariance). *The corrected imputation model trained by $\mathcal{L}_e^{Sup}$ is sufficient for controlling the empirical covariance.*

*(i) For user-item pairs with **observed** outcomes, the empirical covariance is 0. Formally, we have*

$$\left. \frac{\partial}{\partial \epsilon} \mathcal{L}_e^{Sup}(\phi, \epsilon) \right|_{\epsilon = \epsilon^*} = 0, \quad \text{which is equivalent to} \quad \frac{1}{|\mathcal{D}|} \sum_{(u,i):\, o_{u,i}=1} \frac{\hat{p}_{u,i} - o_{u,i}}{\hat{p}_{u,i}} \cdot (e_{u,i} - \tilde{e}_{u,i}) = 0;$$

*(ii) For user-item pairs with **missing** outcomes, suppose that $\hat{p}_{u,i} \ge K_\psi$ and $|e_{u,i} - \tilde{e}_{u,i}| \le K_\phi$, then with probability at least $1 - \eta$, we have*

$$\frac{1}{|\mathcal{D}|} \sum_{(u,i):\, o_{u,i}=0} \frac{\hat{p}_{u,i} - o_{u,i}}{\hat{p}_{u,i}} \cdot (e_{u,i} - \tilde{e}_{u,i})$$

$$\le \underbrace{\sqrt{\mathcal{L}_e^{Sup}(\phi, \epsilon)}}_{\text{proposed loss}} + K_\phi \underbrace{\sqrt{\frac{1}{|\mathcal{D}|} \sum_{u,i \in \mathcal{D}} \left| 1 - \mathbb{E}\left[ \frac{o_{u,i}}{\hat{p}_{u,i}} \Big| x_{u,i} \right] \right|}}_{\text{empirical bias from propensity model}} + \underbrace{\sqrt{K_\phi \left(1 + \frac{1}{K_\psi}\right) \left(2\mathcal{R}(\mathcal{F}) + (2K_\phi + 1)\sqrt{\frac{2\log(4/\eta)}{|\mathcal{D}|}}\right)}}_{\text{tail bound controlled by empirical Rademacher complexity and sample size}},$$

*where the $K_\psi$, $K_\phi$, $\eta$ are constants.*

*Proof.* For the proof of Theorem 4.7(i), first recap that the proposed boosted imputation model is
$$\tilde{e}_{u,i} = m(x_{u,i}; \phi) + \epsilon(o_{u,i} - \pi(x_{u,i}; \psi)),$$
and the proposed corrected imputation loss function for training the boosted imputation model is
$$(\phi^*, \epsilon^*) = \arg\min_{\phi,\epsilon} \mathcal{L}_e^{Sup}(\phi, \epsilon) = \frac{1}{|\mathcal{D}|} \sum_{(u,i)\in\mathcal{D}} \frac{o_{u,i}(e_{u,i} - \tilde{e}_{u,i})^2}{\hat{p}_{u,i}}.$$

By taking the partial derivative with respective to $\epsilon$ of the above formula and setting it to zero, we have
$$\frac{\partial}{\partial\epsilon}\mathcal{L}_e^{Sup}(\phi,\epsilon)\bigg|_{\epsilon=\epsilon^*} = 0, \quad \text{which is equivalent to} \quad \frac{1}{|\mathcal{D}|} \sum_{(u,i):\, o_{u,i}=1} \frac{\hat{p}_{u,i} - o_{u,i}}{\hat{p}_{u,i}} \cdot (e_{u,i} - \tilde{e}_{u,i}) = 0,$$

which proves the empirical convariance on the observed outcomes is 0.

For the proof of Theorem 4.7(ii), by noting that

$$\frac{1}{|\mathcal{D}|} \sum_{(u,i):\, o_{u,i}=0} \frac{\hat{p}_{u,i} - o_{u,i}}{\hat{p}_{u,i}} \cdot (e_{u,i} - \tilde{e}_{u,i}) = \frac{1}{|\mathcal{D}|} \sum_{(u,i):\, o_{u,i}=0} (e_{u,i} - \tilde{e}_{u,i}) \leq \left[ \frac{1}{|\mathcal{D}|} \sum_{(u,i)\in\mathcal{D}} (e_{u,i} - \tilde{e}_{u,i})^2 \right]^{\frac{1}{2}},$$

we now focus on bounding the last term of the above equation with the least probability.

Suppose that $\hat{p}_{u,i} \geq K_\psi$ and $|e_{u,i} - \tilde{e}_{u,i}| \leq K_\phi$, then

$$\frac{1}{|\mathcal{D}|} \sum_{(u,i)\in\mathcal{D}} (e_{u,i} - \tilde{e}_{u,i})^2 = \frac{1}{|\mathcal{D}|} \sum_{(u,i)\in\mathcal{D}} \frac{o_{u,i}(e_{u,i} - \tilde{e}_{u,i})^2}{\hat{p}_{u,i}} + \frac{1}{|\mathcal{D}|} \sum_{(u,i)\in\mathcal{D}} (e_{u,i} - \tilde{e}_{u,i})^2$$

$$- \mathbb{E}\left[ \frac{1}{|\mathcal{D}|} \sum_{(u,i)\in\mathcal{D}} \frac{o_{u,i}(e_{u,i} - \tilde{e}_{u,i})^2}{\hat{p}_{u,i}} \right] + \mathbb{E}\left[ \frac{1}{|\mathcal{D}|} \sum_{(u,i)\in\mathcal{D}} \frac{o_{u,i}(e_{u,i} - \tilde{e}_{u,i})^2}{\hat{p}_{u,i}} \right] - \frac{1}{|\mathcal{D}|} \sum_{(u,i)\in\mathcal{D}} \frac{o_{u,i}(e_{u,i} - \tilde{e}_{u,i})^2}{\hat{p}_{u,i}}$$

$$\leq \frac{1}{|\mathcal{D}|} \sum_{(u,i)\in\mathcal{D}} \frac{o_{u,i}(e_{u,i} - \tilde{e}_{u,i})^2}{\hat{p}_{u,i}} + \left| \frac{1}{|\mathcal{D}|} \sum_{(u,i)\in\mathcal{D}} (e_{u,i} - \tilde{e}_{u,i})^2 - \mathbb{E}\left[ \frac{1}{|\mathcal{D}|} \sum_{(u,i)\in\mathcal{D}} \frac{o_{u,i}(e_{u,i} - \tilde{e}_{u,i})^2}{\hat{p}_{u,i}} \right] \right|$$

$$+ \left( \mathbb{E}\left[ \frac{1}{|\mathcal{D}|} \sum_{(u,i)\in\mathcal{D}} \frac{o_{u,i}(e_{u,i} - \tilde{e}_{u,i})^2}{\hat{p}_{u,i}} \right] - \frac{1}{|\mathcal{D}|} \sum_{(u,i)\in\mathcal{D}} \frac{o_{u,i}(e_{u,i} - \tilde{e}_{u,i})^2}{\hat{p}_{u,i}} \right)$$

$$\leq \mathcal{L}_e^{Sup}(\phi,\epsilon) + K_\phi^2 \cdot \left| \mathbb{E}\left[ \frac{1}{|\mathcal{D}|} \sum_{(u,i)\in\mathcal{D}} 1 - \frac{o_{u,i}}{\hat{p}_{u,i}} \right] \right|$$

$$+ \sup_{f_\theta\in\mathcal{F}} \left( \mathbb{E}\left[ \frac{1}{|\mathcal{D}|} \sum_{(u,i)\in\mathcal{D}} \frac{o_{u,i}(e_{u,i} - \tilde{e}_{u,i})^2}{\hat{p}_{u,i}} \right] - \frac{1}{|\mathcal{D}|} \sum_{(u,i)\in\mathcal{D}} \frac{o_{u,i}(e_{u,i} - \tilde{e}_{u,i})^2}{\hat{p}_{u,i}} \right).$$

For simplicity, we denote the last term in the above formula as

$$\mathcal{B}(\mathcal{F}) = \sup_{f_\theta\in\mathcal{F}} \left( \mathbb{E}\left[ \frac{1}{|\mathcal{D}|} \sum_{(u,i)\in\mathcal{D}} \frac{o_{u,i}(e_{u,i} - \tilde{e}_{u,i})^2}{\hat{p}_{u,i}} \right] - \frac{1}{|\mathcal{D}|} \sum_{(u,i)\in\mathcal{D}} \frac{o_{u,i}(e_{u,i} - \tilde{e}_{u,i})^2}{\hat{p}_{u,i}} \right),$$

we then aim to bound $\mathcal{B}(\mathcal{F})$ in the following.

Note that

$$\mathcal{B}(\mathcal{F}) = \mathbb{E}_{S\sim\mathbb{P}^{|\mathcal{D}|}}[\mathcal{B}(\mathcal{F})] + \left\{ \mathcal{B}(\mathcal{F}) - \mathbb{E}_{S\sim\mathbb{P}^{|\mathcal{D}|}}[\mathcal{B}(\mathcal{F})] \right\},$$

where the first term is $\mathbb{E}_{S\sim\mathbb{P}^{|\mathcal{D}|}}[\mathcal{B}(\mathcal{F})]$, and by Lemma A.1 we have

$$\mathbb{E}_{S\sim\mathbb{P}^{|\mathcal{D}|}}[\mathcal{B}(\mathcal{F})] \leq 2 \mathbb{E}_{S\sim\mathbb{P}^{|\mathcal{D}|}} \mathbb{E}_{\boldsymbol{\sigma}\sim\{-1,+1\}^{|\mathcal{D}|}} \sup_{f_\theta\in\mathcal{F}} \left[ \frac{1}{|\mathcal{D}|} \sum_{(u,i)\in\mathcal{D}} \sigma_{u,i} \frac{o_{u,i}(e_{u,i} - \tilde{e}_{u,i})^2}{\hat{p}_{u,i}} \right].$$

By the assumptions that $\hat{p}_{u,i} \geq K_\psi$ and $|e_{u,i} - \tilde{e}_{u,i}| \leq K_\phi$, we have

$$
\mathop{\mathbb{E}}_{S \sim \mathbb{P}^{|\mathcal{D}|}} [\mathcal{B}(\mathcal{F})] \leq 2K_\phi \left(1 + \frac{1}{K_\psi}\right) \mathop{\mathbb{E}}_{S \sim \mathbb{P}^{|\mathcal{D}|}} \mathbb{E}_{\boldsymbol{\sigma} \sim \{-1,+1\}^{|\mathcal{D}|}} \sup_{f_\theta \in \mathcal{F}} \left[ \frac{1}{|\mathcal{D}|} \sum_{(u,i) \in \mathcal{D}} \sigma_{u,i}(e_{u,i} - \tilde{e}_{u,i}) \right]
$$

$$
= 2K_\phi \left(1 + \frac{1}{K_\psi}\right) \mathop{\mathbb{E}}_{S \sim \mathbb{P}^{|\mathcal{D}|}} \{\mathcal{R}(\mathcal{F})\},
$$

where the last equation is directly from Lemma A.3, and $\mathcal{R}(\mathcal{F})$ is the empirical Rademacher complexity

$$
\mathcal{R}(\mathcal{F}) = \mathbb{E}_{\boldsymbol{\sigma} \sim \{-1,+1\}^{|\mathcal{D}|}} \sup_{f_\theta \in \mathcal{F}} \left[ \frac{1}{|\mathcal{D}|} \sum_{(u,i) \in \mathcal{D}} \sigma_{u,i} e_{u,i} \right],
$$

where $\boldsymbol{\sigma} = \{\sigma_{u,i} : (u,i) \in \mathcal{D}\}$, and $\sigma_{u,i}$ are independent uniform random variables taking values in $\{-1,+1\}$. The random variables $\sigma_{u,i}$ are called Rademacher variables.

By applying McDiarmid's inequality in Lemma A.2, and let $c = \frac{2K_\phi}{|\mathcal{D}|}$, with probability at least $1 - \frac{\eta}{2}$,

$$
\left| \mathcal{R}(\mathcal{F}) - \mathop{\mathbb{E}}_{S \sim \mathbb{P}^{|\mathcal{D}|}} \{\mathcal{R}(\mathcal{F})\} \right| \leq 2K_\phi \sqrt{\frac{\log(4/\eta)}{2|\mathcal{D}|}} = K_\phi \sqrt{\frac{2\log(4/\eta)}{|\mathcal{D}|}}.
$$

For the rest term $\mathcal{B}(\mathcal{F}) - \mathop{\mathbb{E}}_{S \sim \mathbb{P}^{|\mathcal{D}|}} [\mathcal{B}(\mathcal{F})]$, by applying McDiarmid's inequality in Lemma A.2 and the assumptions that $\hat{p}_{u,i} \geq K_\psi$ and $|e_{u,i} - \tilde{e}_{u,i}| \leq K_\phi$, let $c = \frac{2K_\phi^2 \left(1 + \frac{1}{K_\psi}\right)}{|\mathcal{D}|}$, then with probability at least $1 - \frac{\eta}{2}$,

$$
\left| \mathcal{B}(\mathcal{F}) - \mathop{\mathbb{E}}_{S \sim \mathbb{P}^{|\mathcal{D}|}} [\mathcal{B}(\mathcal{F})] \right| \leq 2K_\phi^2 \left(1 + \frac{1}{K_\psi}\right) \sqrt{\frac{\log(4/\eta)}{2|\mathcal{D}|}} = K_\phi^2 \left(1 + \frac{1}{K_\psi}\right) \sqrt{\frac{2\log(4/\eta)}{|\mathcal{D}|}}.
$$

We now bound $\mathcal{B}(\mathcal{F})$ combining the above results. Formally, we have

$$
\mathcal{B}(\mathcal{F}) = \mathop{\mathbb{E}}_{S \sim \mathbb{P}^{|\mathcal{D}|}} [\mathcal{B}(\mathcal{F})] + \left\{ \mathcal{B}(\mathcal{F}) - \mathop{\mathbb{E}}_{S \sim \mathbb{P}^{|\mathcal{D}|}} [\mathcal{B}(\mathcal{F})] \right\}
$$

$$
\leq 2K_\phi \left(1 + \frac{1}{K_\psi}\right) \mathop{\mathbb{E}}_{S \sim \mathbb{P}^{|\mathcal{D}|}} \{\mathcal{R}(\mathcal{F})\} + \left\{ \mathcal{B}(\mathcal{F}) - \mathop{\mathbb{E}}_{S \sim \mathbb{P}^{|\mathcal{D}|}} [\mathcal{B}(\mathcal{F})] \right\}.
$$

With probability at least $1 - \eta$, we have

$$
\mathcal{B}(\mathcal{F}) \leq 2K_\phi \left(1 + \frac{1}{K_\psi}\right) \left( \mathcal{R}(\mathcal{F}) + K_\phi \sqrt{\frac{2\log(4/\eta)}{|\mathcal{D}|}} \right) + K_\phi^2 \left(1 + \frac{1}{K_\psi}\right) \sqrt{\frac{2\log(4/\eta)}{|\mathcal{D}|}}
$$

$$
= K_\phi \left(1 + \frac{1}{K_\psi}\right) \left( 2\mathcal{R}(\mathcal{F}) + K_\phi \sqrt{\frac{18\log(4/\eta)}{|\mathcal{D}|}} \right).
$$

We now bound the empirical convariance on the missing outcomes combining the above results. Formally, we have

$$
\frac{1}{|\mathcal{D}|} \sum_{(u,i):\, o_{u,i}=0} \frac{\hat{p}_{u,i} - o_{u,i}}{\hat{p}_{u,i}} \cdot (e_{u,i} - \tilde{e}_{u,i}) \leq \left[ \frac{1}{|\mathcal{D}|} \sum_{(u,i)\in\mathcal{D}} (e_{u,i} - \tilde{e}_{u,i})^2 \right]^{\frac{1}{2}}
$$

$$
\leq \left[ \mathcal{L}_e^{Sup}(\phi,\epsilon) + \frac{K_\phi^2}{|\mathcal{D}|} \sum_{u,i\in\mathcal{D}} \left| 1 - \mathbb{E}\left[ \frac{o_{u,i}}{\hat{p}_{u,i}} \Big| x_{u,i} \right] \right| + K_\phi \left( 1 + \frac{1}{K_\psi} \right) \left( 2\mathcal{R}(\mathcal{F}) + K_\phi \sqrt{\frac{18\log(4/\eta)}{|\mathcal{D}|}} \right) \right]^{\frac{1}{2}}
$$

$$
\leq \mathcal{L}_e^{Sup}(\phi,\epsilon)^{\frac{1}{2}} + K_\phi \cdot \left[ \frac{1}{|\mathcal{D}|} \sum_{u,i\in\mathcal{D}} \left| 1 - \mathbb{E}\left[ \frac{o_{u,i}}{\hat{p}_{u,i}} \Big| x_{u,i} \right] \right| \right]^{\frac{1}{2}} +
$$

$$
\left[ K_\phi \left( 1 + \frac{1}{K_\psi} \right) \left( 2\mathcal{R}(\mathcal{F}) + K_\phi \sqrt{\frac{18\log(4/\eta)}{|\mathcal{D}|}} \right) \right]^{\frac{1}{2}},
$$

which yields the stated results. □

**Corollary 4.8** (Relation to previous imputed errors). *The learned coefficient $\epsilon^*$ will converge to zero when the imputation model $\hat{e}_{u,i}$ has zero empirical covariance, making $\tilde{e}_{u,i}$ degenerates to $\hat{e}_{u,i}$.*

*Proof of Corollary 4.8.* Note that $\epsilon^*$ is solved by minimizing

$$
\frac{1}{|\mathcal{D}|} \sum_{(u,i)\in\mathcal{D}} \frac{o_{u,i}(e_{u,i} - \hat{e}_{u,i} - \epsilon(o_{u,i} - \hat{p}_{u,i}))^2}{\hat{p}_{u,i}}.
$$

Taking the first derivative of the above loss with respect to $\epsilon$ and setting it to zero yields

$$
\sum_{(u,i)\in\mathcal{D}} \frac{o_{u,i}}{\hat{p}_{u,i}} \cdot \left\{ e_{u,i} - \hat{e}_{u,i} - \epsilon(o_{u,i} - \hat{p}_{u,i}) \right\} \cdot (o_{u,i} - \hat{p}_{u,i}) = 0,
$$

which implies that

$$
\sum_{(u,i)\in\mathcal{D}} \frac{o_{u,i}}{\hat{p}_{u,i}} \cdot \left\{ e_{u,i} - \tilde{e}_{u,i} \right\} \cdot (o_{u,i} - \hat{p}_{u,i}) = 0,
$$

from which implies the uniqueness of $\epsilon$. Formally, if $\hat{e}_{u,i}$ already satisfies zero empirical covariance on the observed outcomes, then $\epsilon = 0$ is a solution of the above equation. Let $\hat{\epsilon}$ be another solution of the above equation. Since the solution of equation is unique, then $\hat{\epsilon}$ will converage to 0, making $\tilde{e}_{u,i}$ degenerates to $\hat{e}_{u,i}$. □

**Corollary 4.9** (Bias reduction property). *The proposed corrected imputation loss leads to the smaller bias of imputed errors $\tilde{e}_{u,i}$, when $\hat{e}_{u,i}$ are inaccurate. Formally, we have*

$$
\min_{\phi,\epsilon} \mathcal{L}_e^{Sup}(\phi,\epsilon) = \frac{1}{|\mathcal{D}|} \sum_{(u,i)\in\mathcal{D}} \frac{o_{u,i}(e_{u,i} - \tilde{e}_{u,i})^2}{\hat{p}_{u,i}} \leq \min_{\phi} \mathcal{L}_e(\phi) = \frac{1}{|\mathcal{D}|} \sum_{(u,i)\in\mathcal{D}} \frac{o_{u,i}(e_{u,i} - \hat{e}_{u,i})^2}{\hat{p}_{u,i}}.
$$

*Proof of Corollary 4.9.* The result holds by noting that

$$
\min_{\phi,\epsilon} \mathcal{L}_e^{Sup}(\phi,\epsilon) \leq \min_{\phi} \mathcal{L}_e^{Sup}(\phi,\epsilon=0) = \min_{\phi} \mathcal{L}_e(\phi) = \frac{1}{|\mathcal{D}|} \sum_{(u,i)\in\mathcal{D}} \frac{o_{u,i}(e_{u,i} - \hat{e}_{u,i})^2}{\hat{p}_{u,i}}.
$$

□

**Corollary 4.10** (Variance reduction property). *The proposed corrected imputation loss leads to the smaller variance of $\tilde{e}_{u,i}$ when the optimal $\epsilon^*$ lies in a certain range. Formally, we have*

$$
\mathbb{V}(\tilde{e}_{u,i}) = \mathbb{V}(\hat{e}_{u,i} + \epsilon^* \cdot (o_{u,i} - \hat{p}_{u,i})) \leq \mathbb{V}(\hat{e}_{u,i}), \;\; if \; \epsilon^* \in \left[ 0, \, 2 \cdot \frac{\mathrm{Cov}(\hat{e}_{u,i}, \hat{p}_{u,i} - o_{u,i})}{\mathbb{V}(\hat{p}_{u,i} - o_{u,i})} \right].
$$

*Proof of Corollary 4.10.* First, we note that $\mathbb{V}(\tilde{e}_{u,i})$ equals to

$$\mathbb{V}(\hat{e}_{u,i}) - 2\epsilon^* \operatorname{Cov}(\hat{e}_{u,i}, \hat{p}_{u,i} - o_{u,i}) + (\epsilon^*)^2 \mathbb{V}(o_{u,i} - \hat{p}_{u,i}),$$

which serves as a quadratic function with respect to $\epsilon^*$. By taking the partial derivative respective to $\epsilon^*$ of the above formula and setting it to zero, the optimal $\epsilon^*$ with the minimal variance is given as

$$\epsilon^* = \frac{\operatorname{Cov}(\hat{e}_{u,i}, \hat{p}_{u,i} - o_{u,i})}{\mathbb{V}(\hat{p}_{u,i} - o_{u,i})}.$$

By exploiting the symmetry of the quadratic function, we have

$$\mathbb{V}(\tilde{e}_{u,i}) = \mathbb{V}(\hat{e}_{u,i} + \epsilon^* \cdot (o_{u,i} - \hat{p}_{u,i})) \leq \mathbb{V}(\hat{e}_{u,i}),$$

$$\text{if} \quad \epsilon^* \in \left[0, \, 2 \cdot \frac{\operatorname{Cov}(\hat{e}_{u,i}, \hat{p}_{u,i} - o_{u,i})}{\mathbb{V}(\hat{p}_{u,i} - o_{u,i})}\right].$$

$\square$

**Theorem 4.11** (Generalization Bound under Superpopulation). *Suppose that $\hat{p}_{u,i} \geq K_\psi$ and $\min\{\tilde{e}_{u,i}, |e_{u,i} - \tilde{e}_{u,i}|\} \leq K_\phi$, then with probability at least $1 - \eta$, we have*

$$\mathcal{L}_{ideal}(\theta) \leq \underbrace{\mathcal{L}_{\text{SuperDR}}(\theta) + \frac{1}{|\mathcal{D}|} \sum_{(u,i)\in\mathcal{D}} \left|1 - \mathbb{E}\left[\frac{o_{u,i}}{\hat{p}_{u,i}}\Big|x_{u,i}\right]\right| \cdot \left|\mathbb{E}[e_{u,i} \mid x_{u,i}] - \mathbb{E}[\tilde{e}_{u,i} \mid x_{u,i}]\right|}_{\text{vanilla DR only controls the empirical DR loss, and empirical risks of imputation and propensity models}}$$

$$+ \underbrace{\left|\frac{1}{|\mathcal{D}|} \sum_{(u,i)\in\mathcal{D}} \operatorname{Cov}\left(\frac{o_{u,i} - \hat{p}_{u,i}}{\hat{p}_{u,i}}, e_{u,i} - \tilde{e}_{u,i}\right)\right|}_{\text{corrected loss further controls the independence}} + \underbrace{\left(1 + \frac{1}{K_\psi}\right)\left(2\mathcal{R}(\mathcal{F}) + K_\phi \sqrt{\frac{18\log(4/\eta)}{|\mathcal{D}|}}\right)}_{\text{tail bound controlled by empirical Rademacher complexity and sample size}}$$

*Proof of Theorem 4.11.* First we decompose the ideal loss as follows.

$$\mathcal{L}_{ideal}(\theta) = \mathcal{L}_{\text{DR}}(\theta) + (\mathcal{L}_{ideal}(\theta) - \mathbb{E}[\mathcal{L}_{\text{DR}}(\theta)]) + (\mathbb{E}[\mathcal{L}_{\text{DR}}(\theta)] - \mathcal{L}_{\text{DR}}(\theta))$$

$$= \mathcal{L}_{\text{DR}}(\theta) + \operatorname{Bias}[\mathcal{L}_{\text{DR}}(\theta)] + (\mathbb{E}[\mathcal{L}_{\text{DR}}(\theta)] - \mathcal{L}_{\text{DR}}(\theta))$$

$$\leq \mathcal{L}_{\text{DR}}(\theta) + |\operatorname{Bias}[\mathcal{L}_{\text{DR}}(\theta)]|$$

$$+ \sup_{f_\theta \in \mathcal{F}} \left(\mathbb{E}\left[\frac{1}{|\mathcal{D}|} \sum_{(u,i)\in\mathcal{D}} \hat{e}_{u,i} + \frac{o_{u,i}(e_{u,i} - \hat{e}_{u,i})}{\hat{p}_{u,i}}\right] - \frac{1}{|\mathcal{D}|} \sum_{(u,i)\in\mathcal{D}} \hat{e}_{u,i} - \frac{o_{u,i}(e_{u,i} - \hat{e}_{u,i})}{\hat{p}_{u,i}}\right).$$

For simplicity, we denote the last term in the above formula as

$$\mathcal{B}(\mathcal{F}) = \sup_{f_\theta \in \mathcal{F}} \left(\mathbb{E}\left[\frac{1}{|\mathcal{D}|} \sum_{(u,i)\in\mathcal{D}} \hat{e}_{u,i} + \frac{o_{u,i}(e_{u,i} - \hat{e}_{u,i})}{\hat{p}_{u,i}}\right] - \frac{1}{|\mathcal{D}|} \sum_{(u,i)\in\mathcal{D}} \hat{e}_{u,i} - \frac{o_{u,i}(e_{u,i} - \hat{e}_{u,i})}{\hat{p}_{u,i}}\right),$$

we then aim to bound $\mathcal{B}(\mathcal{F})$ in the following.

Note that

$$\mathcal{B}(\mathcal{F}) = \mathbb{E}_{S\sim\mathbb{P}^{|\mathcal{D}|}}[\mathcal{B}(\mathcal{F})] + \left\{\mathcal{B}(\mathcal{F}) - \mathbb{E}_{S\sim\mathbb{P}^{|\mathcal{D}|}}[\mathcal{B}(\mathcal{F})]\right\},$$

where the first term is $\mathbb{E}_{S\sim\mathbb{P}^{|\mathcal{D}|}}[\mathcal{B}(\mathcal{F})]$, and by Lemma A.1 we have

$$\mathbb{E}_{S\sim\mathbb{P}^{|\mathcal{D}|}}[\mathcal{B}(\mathcal{F})] \leq 2 \mathbb{E}_{S\sim\mathbb{P}^{|\mathcal{D}|}} \mathbb{E}_{\boldsymbol{\sigma}\sim\{-1,+1\}^{|\mathcal{D}|}} \sup_{f_\theta \in \mathcal{F}} \left[\frac{1}{|\mathcal{D}|} \sum_{(u,i)\in\mathcal{D}} \sigma_{u,i}\hat{e}_{u,i} + \frac{\sigma_{u,i}o_{u,i}(e_{u,i} - \hat{e}_{u,i})}{\hat{p}_{u,i}}\right].$$

By the assumptions that $\hat{p}_{u,i} \geq K_\psi$ and $\min\{\hat{e}_{u,i}, |e_{u,i} - \hat{e}_{u,i}|\} \leq K_\phi$, we have

$$\mathbb{E}_{S\sim\mathbb{P}^{|\mathcal{D}|}}[\mathcal{B}(\mathcal{F})] \leq 2 \mathbb{E}_{S\sim\mathbb{P}^{|\mathcal{D}|}} \mathbb{E}_{\boldsymbol{\sigma}\sim\{-1,+1\}^{|\mathcal{D}|}} \sup_{f_\theta \in \mathcal{F}} \left[\frac{1}{|\mathcal{D}|} \sum_{(u,i)\in\mathcal{D}} \frac{\sigma_{u,i}o_{u,i}(e_{u,i} - \hat{e}_{u,i})}{\hat{p}_{u,i}}\right]$$

$$\leq 2\left(1 + \frac{1}{K_\psi}\right) \mathbb{E}_{S\sim\mathbb{P}^{|\mathcal{D}|}}\{\mathcal{R}(\mathcal{F})\},$$

where the first equation is from Lemma A.3, and $\mathcal{R}(\mathcal{F})$ is the empirical Rademacher complexity

$$\mathcal{R}(\mathcal{F}) = \mathbb{E}_{\boldsymbol{\sigma} \sim \{-1,+1\}^{|\mathcal{D}|}} \sup_{f_\theta \in \mathcal{F}} \left[ \frac{1}{|\mathcal{D}|} \sum_{(u,i) \in \mathcal{D}} \sigma_{u,i} e_{u,i} \right],$$

where $\boldsymbol{\sigma} = \{\sigma_{u,i} : (u,i) \in \mathcal{D}\}$, and $\sigma_{u,i}$ are independent uniform random variables taking values in $\{-1,+1\}$. The random variables $\sigma_{u,i}$ are called Rademacher variables.

By applying McDiarmid's inequality in Lemma A.2, and let $c = \frac{2K_\phi}{|\mathcal{D}|}$, with probability at least $1 - \frac{\eta}{2}$,

$$\left| \mathcal{R}(\mathcal{F}) - \mathbb{E}_{S \sim \mathbb{P}^{|\mathcal{D}|}} \{\mathcal{R}(\mathcal{F})\} \right| \leq 2K_\phi \sqrt{\frac{\log(4/\eta)}{2|\mathcal{D}|}} = K_\phi \sqrt{\frac{2\log(4/\eta)}{|\mathcal{D}|}}.$$

For the rest term $\mathcal{B}(\mathcal{F}) - \mathbb{E}_{S \sim \mathbb{P}^{|\mathcal{D}|}}[\mathcal{B}(\mathcal{F})]$, by applying McDiarmid's inequality in Lemma A.2 and the assumptions that $\hat{p}_{u,i} \geq K_\psi$ and $\min\{\hat{e}_{u,i}, |e_{u,i} - \hat{e}_{u,i}|\} \leq K_\phi$, let $c = \frac{2K_\phi \left(1 + \frac{1}{K_\psi}\right)}{|\mathcal{D}|}$, then with probability at least $1 - \frac{\eta}{2}$,

$$\left| \mathcal{B}(\mathcal{F}) - \mathbb{E}_{S \sim \mathbb{P}^{|\mathcal{D}|}}[\mathcal{B}(\mathcal{F})] \right| \leq 2K_\phi \left(1 + \frac{1}{K_\psi}\right) \sqrt{\frac{\log(4/\eta)}{2|\mathcal{D}|}} = K_\phi \left(1 + \frac{1}{K_\psi}\right) \sqrt{\frac{2\log(4/\eta)}{|\mathcal{D}|}}.$$

We now bound $\mathcal{B}(\mathcal{F})$ combining the above results. Formally, we have

$$\mathcal{B}(\mathcal{F}) = \mathbb{E}_{S \sim \mathbb{P}^{|\mathcal{D}|}}[\mathcal{B}(\mathcal{F})] + \left\{ \mathcal{B}(\mathcal{F}) - \mathbb{E}_{S \sim \mathbb{P}^{|\mathcal{D}|}}[\mathcal{B}(\mathcal{F})] \right\}$$

$$\leq 2\left(1 + \frac{1}{K_\psi}\right) \mathbb{E}_{S \sim \mathbb{P}^{|\mathcal{D}|}}\{\mathcal{R}(\mathcal{F})\} + \left\{ \mathcal{B}(\mathcal{F}) - \mathbb{E}_{S \sim \mathbb{P}^{|\mathcal{D}|}}[\mathcal{B}(\mathcal{F})] \right\}.$$

With probability at least $1 - \eta$, we have

$$\mathcal{B}(\mathcal{F}) \leq 2\left(1 + \frac{1}{K_\psi}\right)\left(\mathcal{R}(\mathcal{F}) + K_\phi \sqrt{\frac{2\log(4/\eta)}{|\mathcal{D}|}}\right) + K_\phi\left(1 + \frac{1}{K_\psi}\right)\sqrt{\frac{2\log(4/\eta)}{|\mathcal{D}|}}$$

$$= \left(1 + \frac{1}{K_\psi}\right)\left(2\mathcal{R}(\mathcal{F}) + K_\phi \sqrt{\frac{18\log(4/\eta)}{|\mathcal{D}|}}\right).$$

We now bound the ideal loss combining the above results. Formally, we have

$$\mathcal{L}_{ideal}(\theta) \leq \mathcal{L}_{\mathrm{DR}}(\theta) + |\mathrm{Bias}[\mathcal{L}_{\mathrm{DR}}(\theta)]| + \mathcal{B}(\mathcal{F})$$

$$\leq \mathcal{L}_{\mathrm{DR}}(\theta) + |\mathrm{Bias}[\mathcal{L}_{\mathrm{DR}}(\theta)]| + \left(1 + \frac{1}{K_\psi}\right)\left(2\mathcal{R}(\mathcal{F}) + K_\phi \sqrt{\frac{18\log(4/\eta)}{|\mathcal{D}|}}\right).$$

In Theorem 4.3, we have already prove that

$$|\mathrm{Bias}[\mathcal{E}_{\mathrm{DR}}(\theta)]| = \left| \frac{1}{|\mathcal{D}|} \sum_{(u,i) \in \mathcal{D}} \mathrm{Cov}\left(\frac{\hat{p}_{u,i} - o_{u,i}}{\hat{p}_{u,i}}, e_{u,i} - \hat{e}_{u,i}\right) \right.$$

$$\left. + \frac{1}{|\mathcal{D}|} \sum_{(u,i) \in \mathcal{D}} \left[ \left\{ 1 - \mathbb{E}\left[\frac{o_{u,i}}{\hat{p}_{u,i}}\Big| x_{u,i}\right] \right\} \cdot \{\mathbb{E}[e_{u,i} \mid x_{u,i}] - \mathbb{E}[\hat{e}_{u,i} \mid x_{u,i}]\} \right] \right|$$

$$\leq \left| \frac{1}{|\mathcal{D}|} \sum_{(u,i) \in \mathcal{D}} \mathrm{Cov}\left(\frac{o_{u,i} - \hat{p}_{u,i}}{\hat{p}_{u,i}}, e_{u,i} - \hat{e}_{u,i}\right) \right|$$

$$+ \frac{1}{|\mathcal{D}|} \sum_{(u,i) \in \mathcal{D}} \left| 1 - \mathbb{E}\left[\frac{o_{u,i}}{\hat{p}_{u,i}}\Big| x_{u,i}\right] \right| \cdot \left| \mathbb{E}[e_{u,i} \mid x_{u,i}] - \mathbb{E}[\hat{e}_{u,i} \mid x_{u,i}] \right|,$$

therefore with probability at least $1 - \eta$, we have

$$\mathcal{L}_{ideal}(\theta) \le \mathcal{L}_{\mathrm{DR}}(\theta) + \frac{1}{|\mathcal{D}|} \sum_{(u,i) \in \mathcal{D}} \left| 1 - \mathbb{E}\left[ \frac{o_{u,i}}{\hat{p}_{u,i}} | x_{u,i} \right] \right| \cdot \left| \mathbb{E}[e_{u,i} \mid x_{u,i}] - \mathbb{E}[\hat{e}_{u,i} \mid x_{u,i}] \right|$$

$$+ \left| \frac{1}{|\mathcal{D}|} \sum_{(u,i) \in \mathcal{D}} \mathrm{Cov}\left( \frac{o_{u,i} - \hat{p}_{u,i}}{\hat{p}_{u,i}}, e_{u,i} - \hat{e}_{u,i} \right) \right| + \left( 1 + \frac{1}{K_\psi} \right) \left( 2\mathcal{R}(\mathcal{F}) + K_\phi \sqrt{\frac{18 \log(4/\eta)}{|\mathcal{D}|}} \right),$$

which yields the stated results. $\qquad \square$

