# OpenReview forum: "Unveiling Extraneous Sampling Bias with Data Missing-Not-At-Random"
_NeurIPS.cc/2025/Conference — NeurIPS 2025 poster_

### Official Review · Reviewer_FSqV · 2025-06-20

**Clarity:** 2
**Significance:** 2
**Originality:** 3
**Rating:** 4
**Confidence:** 3

**Summary:**

The paper identifies a critical limitation in existing debiasing methods, which assume "exact matching" between training and test sets. It introduces a more realistic "super-population" setting, where samples are drawn independently from the same underlying distribution. Under this setting, even perfectly specified DR estimators are biased due to a new covariance term. To address this, the authors propose SuperDR, which corrects the imputation model by introducing a learnable parameter ε to minimize the empirical covariance, inspired by TMLE.

**Questions:**

* Was any propensity score stabilization used?
* How were the inner loop steps and learning rates chosen in Algorithm 1?
* Why apply correction to the imputation model rather than the prediction output?
* Are MF-based features sufficient for generalization across disjoint user-item samples?

**Ethical Concerns:**

["NO or VERY MINOR ethics concerns only"]

**Final Justification:**

My concerns have been adequately addressed, and I have decided to maintain my positive rating.

**Limitations:**

Yes

**Quality:**

3

**Strengths And Weaknesses:**

#### **Strengths:**

* The clear distinction between exact matching, cold-start, and the proposed super-population setting fills an important gap and better reflects real-world applications.

* The paper formally derives the bias and shows how the correction term effectively targets the covariance source. Theoretical results (e.g., Theorem 4.3, 4.7) are elegant and insightful.

*  Experiments on multiple datasets, including the large-scale KuaiRec, demonstrate significant and consistent improvements. Analyses directly validate theoretical claims (e.g., performance vs. empirical covariance).

**Weaknesses:**

*  Lack of detail on whether stabilization techniques (e.g., propensity clipping) were used, which affects reproducibility and variance control explanation.
*  The alternating optimization scheme increases overhead but is not analyzed or discussed in terms of cost-performance trade-offs.
*  Lack of some important related work:
[1] Uncertainty Calibration for Counterfactual Propensity Estimation in Recommendation
[2] Cdr:Conservative doubly robust learning for debiased recommendation

---

> ### Author Rebuttal · Authors · 2025-07-31
>
> We thank the reviewer for efforts in evaluating our work and helpful suggestions. Below, we will try our best to address your concerns and questions.
>
> > **W1 & Q1:** Lack of detail on whether stabilization techniques (e.g., propensity clipping) were used, which affects reproducibility and variance control explanation.
>
> **Response to W1 & Q1:** We thank the reviewer for pointing out these issues. Yes, we use a propensity clipping technique, and the clipping threshold is a hyperparameter tuned in [0.005, 0.05] on all three datasets. The best threshold is 0.0186 for **Coat**, 0.0127 for **Yahoo! R3**, and 0.0112 for the **KuaiRec** dataset. We will add these details in our revised version.
>
> > **W2:** The alternating optimization scheme increases overhead but is not analyzed or discussed in terms of cost-performance trade-offs.
>
> **Response to W2:** We thank the reviewer for raising this meaningful concern.
> - Note that the proposed method subtly modifies the imputation loss $\hat e$ by adding $\epsilon (o-p)$, which **theoretically does not increase complexity compared to the original joint learning method, due to we only require one more gradient calculation and backward propagation for a single learnable parameter $\epsilon$.**
> - Empirically, the model structurally is also not more complex. Besides a single learnable parameter $\epsilon$, no new learnable parameters are introduced.
> - **We also add training time experiments on all three datasets under varying dataset scales (controlled by the ratio parameter $b$, varying in {0.1, 0.3, 0.5, 0.7, 0.9}, i.e., we randomly sample 100b% of users and items).** We take DR-JL as the baseline. Thus, all results are 1 for DR-JL. The results show that the time complexity of our method is close to the doubly robust joint learning method, and much less than the previous SOTA DR-based methods, demonstrating the efficiency of our method.
>
> |**Coat**|b=0.1|b=0.3|b=0.5|b=0.7|b=0.9|
> |-|-|-|-|-|-|
> |DR-JL|1.000|1.000|1.000|1.000|1.000|
> |DR-MSE|1.435|1.956|2.008|1.619|1.183|
> |MR|1.588|2.206|2.365|1.524|1.296|
> |DR-V2|1.298|2.093|1.850|1.405|1.028|
> |DCE-TDR|2.235|2.906|3.252|1.912|1.596|
> |Ours|**1.078**|**1.122**|**1.273**|**1.185**|**0.956**|
> |**Yahoo! R3**||||||
> |DR-JL|1.000|1.000|1.000|1.000|1.000|
> |DR-MSE|1.831|2.583|2.866|3.401|2.808|
> |MR|**1.253**|2.458|2.907|2.689|2.862|
> |DR-V2|1.429|1.958|**1.320**|1.921|1.891|
> |DCE-TDR|2.014|3.271|3.247|2.864|3.075|
> |Ours|1.255|**1.401**|1.536|**1.424**|**1.356**|
> |**KuaiRec**||||||
> |DR-JL|1.000|1.000|1.000|1.000|1.000|
> |DR-MSE|1.371|1.706|2.871|1.867|2.714|
> |MR|1.604|2.294|3.355|2.036|3.036|
> |DR-V2|1.266|1.647|1.677|1.012|2.631|
> |DCE-TDR|2.734|4.000|5.065|2.831|3.607|
> |Ours|**1.354**|**1.412**|**1.129**|**0.827**|**1.726**|
> |
>
> > **W3:** Lack of some important related work: [1] Uncertainty Calibration for Counterfactual Propensity Estimation in Recommendation [2] Cdr:Conservative doubly robust learning for debiased recommendation.
>
> **Response to W3:** We thank the reviewer for the useful suggestions. We will modify the related work section to include these two important works. In addition, we add experiments to compare our method to these two methods. The results are shown below:
>
> |Methods|**Coat**|||**Yahoo! R3**|||**KuaiRec**|||
> |-|-|-|-|-|-|-|-|-|-|
> ||AUC|NDCG@5|Recall@5|AUC|NDCG@5|Recall@5|AUC|NDCG@50|Recall@50|
> |DR-JL|0.733±0.003|0.650±0.005|0.625±0.007|0.667±0.005|0.655±0.004|0.449±0.008|0.818±0.003|0.620±0.004|0.655±0.007|
> |CDR [2] |0.743±0.004|0.652±0.005|0.627±0.005|0.672±0.004|0.658±0.003|0.454±0.005|0.823±0.005|0.623±0.004|0.660±0.005|
> |MRDR-CAL [1] |0.744±0.003|0.651±0.004|0.630±0.004|0.675±0.003|0.660±0.003|0.454±0.005|0.822±0.006|0.625±0.005|0.665±0.006|
> |DCE-TDR|0.746±0.005|0.654±0.005|0.629±0.006|0.679±0.004|0.662±0.005|0.459±0.004|0.824±0.003|0.632±0.004|0.659±0.004|
> |D-DR|0.750±0.004|0.654±0.004|0.630±0.005|0.678±0.004|0.659±0.004|0.456±0.003|0.822±0.004|0.630±0.005|0.672±0.005|
> |Ours|**0.757±0.004**|**0.667±0.005**|**0.637±0.007**|**0.686±0.003**|**0.667±0.004**|**0.463±0.003**|**0.828±0.004**|**0.640±0.005**|**0.680±0.005**|
>
> We find that our method still outperforms the baselines.
>
> > **Q2:** How were the inner loop steps and learning rates chosen in Algorithm 1?
>
> **Response to Q2:**  We thank the reviewer for the question. Following the previous implementation in [1-3], the step is fixed to 1. In addition, as shown in lines 250-251 in the manuscript, the learning rate is tuned in {0.001, 0.005, 0.01, 0.02, 0.05} for parameters in prediction, imputation, and propensity model, and in {0.01, 0.05, 0.1, 0.15, 0.2} for $\epsilon$. The best choice is {0.01, 0.15} for **Coat**, {0.005, 0.15} for **Yahoo! R3**, and {0.005, 0.1} for **KuaiRec**.
>
> > **Q3:** Why apply correction to the imputation model rather than the prediction output?
>
> **Response to Q3:**   We thank the reviewer for the interesting question. **The conclusion is, applying the correction to either the imputation model or the prediction model will lead to exactly the same results!** Here are the reasons.
> - For updating the prediction model, we first recap the vanilla DR loss below:
>
>     $$\mathcal L_{\text{DR}}(\theta)=\frac{1}{|\mathcal{D}|} \sum_{(u, i) \in \mathcal{D}}\left[\hat e_{u, i}+\frac{o_{u, i}\left(e_{u, i}-\hat e_{u, i}\right)}{\hat p_{u, i}}\right].$$
> **For updating the prediction model, note that whether adding the correction to the imputation model or the prediction model based on this DR loss will not affect the updating of the prediction model, because there is no gradient to the prediction model.**
>
> - For updating the imputation model, we recap the vanilla imputation loss function below
>     $$\mathcal L_e = \frac{1}{|\mathcal{D}|} \sum_{(u, i) \in \mathcal{D}} \frac{o_{u, i}\left(e_{u, i}-\hat e_{u, i}\right)^2}{\hat p_{u, i}}.$$
>
> **Note that adding the correction to the imputation model or the prediction model based on this imputation loss also makes no difference when updating the imputation model and the parameter $\epsilon$.**
>
> > **Q4:** Are MF-based features sufficient for generalization across disjoint user-item samples?
>
> **Response to Q4:**
>
> We thank the reviewer for raising this concern.
>
> - **We add experiments on all three datasets, demonstrating that MF-based features are sufficient for generalization.** Specifically, for all three datasets, we randomly split the training data (i.e., MNAR data) into training / validation / test data, with a ratio 0.8 / 0.1 / 0.1. Then we use an MLP as the backbone to train a prediction model using the naive loss, i.e., loss on the observed user-item pairs.
> $$\mathcal L_{\text{naive}}(\theta)=\frac{1}{|\mathcal{O}|} \sum_{(u, i) \in \mathcal{O}}e_{u,i}.$$
>
> The test performance on all three datasets corresponding to MSE and AUC is shown below:
> |Datasets|AUC|MSE|
> |-|-|-|
> |**Coat**|0.952|0.092|
> |**Yahoo! R3**|0.946|0.096|
> |**KuaiRec**|0.889|0.117|
>
> - The results show that simple naive loss-based neural networks demonstrate excellent performance without data MNAR on all three datasets, proving MF-based features are sufficient for generalization.
> - Furthermore, since **KuaiRec** is a more complicated industrial dataset, its lower results compared to **Coat** and **Yahoo! R3** are in line with expectations.
> - Finally, naive methods perform poorly with data MNAR (**Coat** and **Yahoo! R3** have around 20% lower AUC, and KuaiRec around 10% lower AUC), highlighting the need for debiasing in real-world datasets.
>
> **We believe the above discussion has thoroughly addressed your concerns regarding our work. We would be grateful if you could kindly consider raising your rating.** Thank you!

---

> ### Comment · Reviewer_FSqV · 2025-08-08
>
> Thank you for your detailed response. My concerns have been adequately addressed, and I have decided to maintain my positive rating.

---

> > ### Author Response · Authors · 2025-08-08
> >
> > We are happy to hear that your concerns have been adequately addressed. We are truly grateful for your dedicated time and constructive comments!

---

### Official Review · Reviewer_gNuC · 2025-06-25

**Clarity:** 2
**Significance:** 3
**Originality:** 3
**Rating:** 3
**Confidence:** 2

**Summary:**

This paper investigates a practical and underexplored challenge in unbiased learning: selection bias in scenarios where training and test sets do not share identical units (e.g., different users/items in recommender systems), though the joint distribution of features and outcomes remains the same. The authors argue that existing doubly robust (DR) estimators, which assume exact matching, are insufficient in this "super-population" setting. They derive the theoretical bias of DR under this new scenario and identify a non-trivial covariance term that induces additional bias. To address this, the paper proposes SuperDR, a novel estimator with a corrected imputation model designed to control this covariance term. They provide theoretical guarantees on bias, variance, and generalization error, and validate their method on three real-world datasets, demonstrating superior performance over existing approaches.

**Questions:**

Please refer to the Weaknesses.

**Ethical Concerns:**

["NO or VERY MINOR ethics concerns only"]

**Limitations:**

Yes.

**Quality:**

2

**Strengths And Weaknesses:**

**Strengths**

**S1. Principled Solution**: The proposed SuperDR method is well-motivated and theoretically grounded. The use of imputation correction inspired by targeted learning is elegant and practical.

**S2. Strong Empirical Results**: Experiments across three datasets (Coat, Yahoo! R3, KuaiRec) demonstrate that SuperDR consistently outperforms strong DR baselines, particularly in low-overlap scenarios between train and test data.

**Weaknesses**

**W1. Assumption of Equal Joint Distributions:** The super-population scenario assumes $P_{\rm train}(x,r)=P_{\rm test}(x,r)$; in practice, covariate drift or nonstationarity could violate this and undermine unbiasedness.

**W2. Dependence on Propensity Pre-training:** Requires a pre-trained propensity model (e.g., logistic regression) whose miscalibration could propagate through the correction step, yet sensitivity to propensity errors is not deeply explored.

**W3. Computational Overhead:** Jointly learning three components (prediction, imputation, and ε) may increase training time and tuning complexity compared to standard DR methods; scalability to very large item × user spaces warrants further study.

**W4. Limited Domain Scope:** Experiments focus on recommender-system datasets; applicability to other MNAR settings (e.g., causal inference, medical missingness) remains to be demonstrated.

---

> ### Author Rebuttal · Authors · 2025-07-31
>
> We thank the reviewer for efforts in evaluating our work and helpful suggestions. Below, we will try our best to address your concerns and questions.
>
> > **W1. Assumption of Equal Joint Distributions:**   The super-population scenario assumes $P_{\rm train}(x,r)=P_{\rm test}(x,r)$; in practice, covariate drift or nonstationarity could violate this and undermine unbiasedness.
>
> **Response to W1:** We thank the reviewer for the question. **However, we think there is a huge misunderstanding.**
> - Note that we assume $P_{\rm train}(x,r)=P_{\rm test}(x,r)$ means that the target data distribution (not the observed data distribution) in training is the same as the distribution in test data. Then, due to the MNAR issue, the observed data distribution $P_{\rm obs}(x,r)$ is different from the test data.
> -  For example, the Coat dataset includes 290 users and 300 items with 6,960 MNAR observations, then **we only assume target data distribution in training $P_{\rm train}(x,r)$ (the distribution of 290 * 300 = 87,000 user-item pairs) the same as the test distribution $P_{\rm test}(x,r)$, and we do not assume distribution on 6,960 observations $P_{\rm obs}(x,r)$ the same as the test distribution.** We will make it clearer in our revised manuscript.
>
> > **W2. Dependence on Propensity Pre-training:** Requires a pre-trained propensity model (e.g., logistic regression) whose miscalibration could propagate through the correction step, yet sensitivity to propensity errors is not deeply explored.
>
> **Response to W2:** We thank the reviewer for the question. The existing DR-based training algorithms are mainly divided into two types:
>
> - The first is based on the joint training algorithm, which pretrains the propensity model, and then trains the imputation model and the prediction model, such as DR-JL [1], MRDR-JL [2], DCE-DR [3], etc.
> - The second is based on the multi-task learning algorithm, which learns the propensity model, the prediction model, and the imputation model together, such as ESCM2-DR [4], SDR [5], DR-V2 [6], etc.
>
> As suggested by the reviewer, **we added experiments on all three datasets, Coat, Yahoo, and Kuairec, where the propensity model is 1 layer (i.e., logistic regression), 2 layers, and 3 layers neural networks, respectively, to explore the sensitivity to propensity errors of our method. In addition, we also implemented both joint learning and multi-task learning as the learning algorithm to investigate whether our method needs a pre-trained propensity model.** The results are as follows:
>
> |Num of layers in propensity model|**Coat**|||**Yahoo! R3**|||**KuaiRec**|||
> |-|-|-|-|-|-|-|-|-|-|
> ||AUC|NDCG@5|Recall@5|AUC|NDCG@5|Recall@5|AUC|NDCG@50|Recall@50|
> |Layer1_MTL|0.754|0.663|0.638|**0.688**|**0.669**|**0.467**|0.829|0.633|0.670|
> |Layer1_JL|0.753|0.661|**0.649**|0.674|0.654|0.450|0.826|0.631|0.667|
> |Layer2_MTL|0.752|0.663|0.636|**0.688**|0.668|0.464|0.822|0.637|0.674|
> |Layer2_JL|**0.757**|0.667|0.637|0.686|0.667|0.463|0.828|0.640|0.680|
> |Layer3_MTL|0.754|**0.669**|0.638|0.678|0.659|0.455|**0.830**|**0.641**|**0.681**|
> |Layer3_JL|0.740|0.655|0.624|0.684|0.665|0.464|0.829|0.636|0.675|
> |Best baseline|0.750|0.654|0.630|0.679|0.662|0.459|0.824|0.632|0.671|
> |
>
> The results show that our method is stable to the choice of propensity model and learning algorithm.
>
> > **W3. Computational Overhead:** Jointly learning three components (prediction, imputation, and $\epsilon$) may increase training time and tuning complexity compared to standard DR methods; scalability to very large item × user spaces warrants further study.
>
> **Response to W3:** We thank the reviewer for raising this meaningful concern.
> - Note that the proposed method subtly modifies the imputation loss $\hat e$ by adding $\epsilon (o-p)$, which **theoretically does not increase complexity compared to the original joint learning method, due to we only require one more gradient calculation and backward propagation for a single learnable parameter $\epsilon$.**
> - Empirically, the model structurally is also not more complex. Besides a single learnable parameter $\epsilon$, no new learnable parameters are introduced.
> - **We also add training time experiments on all three datasets under varying dataset scales (controlled by the ratio parameter $b$, varying in {0.1, 0.3, 0.5, 0.7, 0.9}, i.e., we randomly sample 100b% of users and items).** We take DR-JL as the baseline. Thus, all results are 1 for DR-JL. The results show that the time complexity of our method is close to the doubly robust joint learning method, and much less than the previous SOTA DR-based methods, demonstrating the efficiency of our method.
>
> |**Coat**|b=0.1|b=0.3|b=0.5|b=0.7|b=0.9|
> |-|-|-|-|-|-|
> |DR-JL|1.000|1.000|1.000|1.000|1.000|
> |DR-MSE|1.435|1.956|2.008|1.619|1.183|
> |MR|1.588|2.206|2.365|1.524|1.296|
> |DR-V2|1.298|2.093|1.850|1.405|1.028|
> |DCE-TDR|2.235|2.906|3.252|1.912|1.596|
> |Ours|**1.078**|**1.122**|**1.273**|**1.185**|**0.956**|
> |**Yahoo! R3**||||||
> |DR-JL|1.000|1.000|1.000|1.000|1.000|
> |DR-MSE|1.831|2.583|2.866|3.401|2.808|
> |MR|**1.253**|2.458|2.907|2.689|2.862|
> |DR-V2|1.429|1.958|**1.320**|1.921|1.891|
> |DCE-TDR|2.014|3.271|3.247|2.864|3.075|
> |Ours|1.255|**1.401**|1.536|**1.424**|**1.356**|
> |**KuaiRec**||||||
> |DR-JL|1.000|1.000|1.000|1.000|1.000|
> |DR-MSE|1.371|1.706|2.871|1.867|2.714|
> |MR|1.604|2.294|3.355|2.036|3.036|
> |DR-V2|1.266|1.647|1.677|1.012|2.631|
> |DCE-TDR|2.734|4.000|5.065|2.831|3.607|
> |Ours|**1.354**|**1.412**|**1.129**|**0.827**|**1.726**|
> |
>
> > **W4. Limited Domain Scope:** Experiments focus on recommender-system datasets; applicability to other MNAR settings (e.g., causal inference, medical missingness) remains to be demonstrated.
>
> **Response to W4:** We thank the reviewer for the question. **As suggested by the reviewer, we further explore the effectiveness of our method on two scenarios, the medical field and business field, with two datasets, IHDP and Jobs.**
> - The semi-synthetic IHDP dataset [7] is constructed from the Infant Health and Development Program (IHDP) with 747 individuals and 25 covariates.
> - The real-world Jobs dataset [8] is based on the National Supported Work program with 3,212 individuals and 17 covariates.
>
> In this scenario, the DR estimator has the following form:
>
> $$\tau(x)=E[\frac{T(Y-m_1(X))}{\pi(X)}-\frac{(1-T)(Y-m_0(X))}{1-\pi(X)}+m_1(X)-m_0(X) | X=x],$$
>
> where $T \in $ {0,1} is the binary treatment indicator, $Y$ is the outcome, $\pi(X) = P(T=1|X)$ is the propensity model, and $m_0(X), m_1(X)$ are the imputation models for $T = 0$ and $T = 1$, respectively.
>
> Following the previous study [9], we split the data into training/validation/testing sets with ratios 63/27/10 and 56/24/20, with 10 repeated times on the IHDP and the Jobs datasets, respectively. **Note that each sample is already disjoint, and the training distribution is the same as the test distribution due to we randomly split the data.** For evaluation, we use  $\sqrt{\epsilon_{\text{PEHE}}}$ and  $\epsilon_{\text{ATE}}$ for IHDP, and  $R_{\text{pol}}$ and  $\epsilon_{\text{ATT}}$ for Jobs. The definitions are shown below:
>
> - $\epsilon_{\text{PEHE}} =
> {\frac{1}{N} \sum_{i=1}^N [(\hat {Y}_i(1) - \hat {Y}_i(0)) - (Y_i(1) - Y_i(0))]^2}$, where $\hat {Y}_i(1)$ and $\hat {Y}_i(0)$ are the predicted values for the corresponding true potential outcomes of unit $i$.
> - $\epsilon_{\text{ATE}}=\frac{1}{N}| \sum_{i=1}^N ((\hat {Y}_i(1) - \hat {Y}_i(0)) - (Y_i(1) - Y_i(0)))|.$
> - ${R}_{\text{Pol}}= 1-(\mathbb{E}[Y(1) \mid \hat {Y}(1) - \hat {Y}(0) >0, X=1] \cdot \mathbb{P}(\hat {Y}(1) - \hat {Y}(0)>0)+\mathbb{E}[Y(0) \mid \hat {Y}(1) - \hat {Y}(0) \leq 0, X=0] \cdot \mathbb{P}(\hat {Y}(1) - \hat {Y}(0) \leq 0))$, where $T, C, E$ are the indexes of treatment, control, and randomized sample set, respectively.
> -  $\epsilon_{\text{ATT}} = |\text{ATT} - \frac{1}{|T|} \sum_{i \in T} (\hat {Y}_i(1) - \hat {Y}_i(0)|$
>
> where $\text{ATT} = |\frac{1}{|T|} \sum_{i \in T}{Y_i} - \frac{1}{|C \cap E|} \sum_{i \in (C \cap E)}{Y_i}|$.
>
> The lower the metrics, the better the performance. See [9] for a more detailed definition of evaluation metrics. The performance on the test set is shown below:
>
> Results on IHDP:
> |Methods|$\sqrt{\epsilon_{\text{PEHE}}}$↓|$\epsilon_{\text{ATE}}$↓|
> |-|-|-|
> |DR-JL|1.37±0.30|0.22±0.15|
> |Ours|**0.95±0.11**|**0.14±0.09**|
> |
>
> Results on Jobs:
> |Methods|${R}_{\text{Pol}}$↓|$\epsilon_{\text{ATT}}$↓|
> |-|-|-|
> |DR-JL|0.27±0.06|**0.05±0.03**|
> |Ours|**0.23±0.07**|**0.05±0.02**|
> |
>
> The results show that our method can be applied to those datasets, and the performance is better than the DR-JL method.
>
> **We believe the above discussion has thoroughly addressed your concerns regarding our work. We would be grateful if you could kindly consider raising your rating.** Thank you!
>
> ---
> References
>
> [1] Doubly Robust Joint Learning for Recommendation on Data Missing Not at Random. ICML 2019.
>
> [2] Enhanced Doubly Robust Learning for Debiasing Post-Click Conversion Rate Estimation. SIGIR 2021.
>
> [3] Doubly Calibrated Estimator for Recommendation on Data Missing Not At Random. WWW 2024.
>
> [4] ESCM$^2$: Entire Space Counterfactual Multi-Task Model for Post-Click Conversion Rate Estimation. SIGIR 2022.
>
> [5] StableDR: Stabilized Doubly Robust Learning for Recommendation on Data Missing Not at Random. ICLR 2023.
>
> [6] Propensity Matters: Measuring and Enhancing Balancing for Recommendation. ICML 2023.
>
> [7] Bayesian Nonparametric Modeling for Causal Inference. Journal of Computational and Graphical Statistics 2011.
>
> [8] Evaluating the Econometric Evaluations of Training Programs with Experimental Data. The American Economic Review 1986.
>
> [9] Estimating Individual Treatment Effect: Generalization Bounds and Algorithms. ICML 2017.

---

> ### Author Response · Authors · 2025-08-03
> **We would like to supplement more experiments and discussion on "W1. Assumption of Equal Joint Distributions" to fully address your concerns!**
>
> Dear Reviewer gNuC,
>
> Thanks again for your insightful comments and interesting questions. During the rebuttal period, we have validated time efficiency, insensitivity to the propensity model, and applicability to other domains (such as medical and business) of our method. In addition, we highlight that the previous method requires “exact match user and item” between the training set and test set, while our method only assumes $P_{\rm train}(x,r)=P_{\rm test}(x,r)$. **To further validate our method's applicability in real-world scenarios (may have covariate shift or nonstationarity), we show our method is still valid and outperforms baselines if the assumption $P_{\rm train}(x,r)=P_{\rm test}(x,r)$ is violated.**
>
> We add experiments on the industrial dataset **KuaiRec**. To process the data, we first randomly split 80% users and items to construct the training set to make sure we do not see all users and items during training. Then we consider the scenario with $P_{\rm train}(x,r) \neq P_{\rm test}(x,r)$:
>
> - We assign a propensity according to the item frequency and the user's active level (measured by the number of rated items) to split the dataset. This will lead to the first training set containing popular users and items, and the second containing non-popular users and items. We set the threshold to make the number of user-item pairs the same in the two datasets.
>
> For all methods, we test on the MAR dataset. The results are shown below:
>
> |||AUC|Recall@50|NDCG@50|
> |-|-|-|-|-|
> |First set (popular users & items)|DR-JL|0.819|0.670|0.633|
> ||Ours|**0.823**|**0.666**|**0.629**|
> |Second set (non-popular users & items)|DR-JL|0.705|0.480|0.478|
> ||Ours|**0.717**|**0.502**|**0.507**|
> |
>
> **The results show that our method still outperforms the baseline DR-JL, especially under the second set (when the covariate shift is large). One possible reason is that our method deals with a super-population scenario, which is more robust toward a huge distribution shift.**
>
>
> A more practical scenario is that **data may arrive sequentially with $P_{\rm train}(x,r) \neq P_{\rm test}(x,r)$, and we need to retrain our model efficiently on new data.** To further show the applicability of our method, we evaluate whether our method can be efficiently applied in this scenario. We propose two ways to update our predictions efficiently:
> - We store the parameters in models and optimizers at the end of training, and fine-tune them when new data comes (named "continue training").
> - We freeze the parameters in prediction, imputation, and propensity model, and add a learnable shallow residual structure, which is zero-initialized when new data comes (named "adding residual").
> - Also, we consider the full retraining of all parameters on both old and new data as the oracle baselines.
>
> We also add experiments on the industrial dataset **KuaiRec** and first randomly split 80% users and items. Then we consider the following scenario:
> - We randomly split the user set and item set into two parts with a ratio of 50/50, and train our method on the first dataset. Then we assign a propensity according to the item frequency and the user's active level, sampling popular users and items into the first time retraining, and the remaining into the second time retraining. Also, we adjust the threshold to make the number of user-item pairs the same in each retraining.
>
> The results on the MAR test data are shown below:
> |Data||Inconsistent in distribution|||
> |-|-|-|-|-|
> |||AUC|NDCG@50|Recall@50|AUC|NDCG@50|Recall@50|
> |Train Set1 (50%)|DR-JL|0.817|0.623|0.663|
> ||Ours|**0.825**|**0.636**|**0.678**|
> |Train Set2 (25%)|Continue training on DR-JL|0.811|0.631|0.666|
> ||Continue training on ours|**0.821**|**0.633**|**0.669**|
> ||Adding residual on DR-JL|0.816|0.620|0.653|
> ||Adding residual on ours|**0.821**|**0.640**|**0.680**|
> ||Full training on DR-JL|0.821|0.626|0.665|
> ||Full training on ours|**0.827**|**0.640**|**0.679**|
> |Train Set3 (25%)|Continue training on DR-JL|0.712|0.457|0.449|
> ||Continue training on ours|**0.729**|**0.492**|**0.504**|
> ||Adding residual on DR-JL|0.715|0.450|0.449|
> ||Adding residual on ours|**0.725**|**0.479**|**0.506**|
> ||Full training on DR-JL|0.822|0.627|0.668| |
> ||Full training on ours|**0.828**|**0.640**|**0.679**|
>
> - We find that our methods can efficiently adapt our predictions as new data becomes available.
> - In addition, when there is a large gap between the new data and the original training data (such as non-popular users and items, as shown in the "Train Set3”), though the performance of our method will drop, our method significantly outperforms DR-JL.
>
> **By now, we hope we have fully addressed your concerns! We sincerely expect that the added extensive experiments and discussions on time efficiency, insensitivity to the propensity model, and applicability to other domains (such as medical and business) as well as the assumption $P_{\rm train}(x,r)=P_{\rm test}(x,r)$ is violated, will enable you to raise your rating!**

---

> > ### Comment · Reviewer_gNuC · 2025-08-04
> >
> > Thanks for the rebuttal. However, the explanations regarding W1 and W2 are still unclear.
> >
> > For W1, first and foremost, the covariate drift or nonstationarity of data distribution means the preference shift of users or items becoming outdated, rather than the observed data. Second, the rebuttal contains many elusive concepts. For example, what is 'target data distribution in training distribution' ? Why is there a distribution in a distribution? what is 'popular user'? I assume the authors use this term to mean 'active user'. Besides, the results on the first set (popular users & items) of KuaiRec show DR-JL outperforms the proposed method w.r.t. Recall and NDCG. In addition, it is elusive why testing on popular and unpopular items separately can demonstrate that the proposed method can handle nonstationarity between training and test sets. Last but not least, what is the necessity of introducing a new baseline, DR-JL?
> >
> > For W2, my concern is that the errors made by the propensity model may degrade the final accuracy. However, the authors compare two types of methods, both of which involve a propensity model, and the results show that these two types of methods achieve comparable accuracy. I don't understand how such results can address my concern.
> >
> > Besides, the above rebuttals are merely built upon experimental results, which are not sufficiently convincing.

---

> > > ### Author Response · Authors · 2025-08-07
> > >
> > > Thank you for your thoughtful review and valuable questions! We address your questions point-to-point in the following.
> > >
> > > > For W1, first and foremost, the covariate drift or nonstationarity of data distribution means the preference shift of users or items becoming outdated, rather than the observed data.
> > >
> > > Thank you for the great question! We would like to clarify that **covariate drift or nonstationarity of data distribution is _orthogonal_ to our studied problem.**
> > >
> > > - We fully agree that the covariate drift or nonstationarity of data distribution means the preference shift of users or items becoming outdated, rather than the observed data, which is widely exist and has been well studied in the out-of-distribution (OOD) community, e.g., OOD in recommender system [1].
> > >
> > > - However, our paper focus on the sample selection bias, which is formally equivalent to the semi-supervised learning (SSL) with missing-not-at-random (MNAR) labels. In SSL articles, people are more concerned with covariate drift $P_{obs}(X, Y)=P(X, Y\mid O=1)\neq P(X, Y)$ due to MNAR labeled data, in which $O$ is the labeled data indicator. They usually assume that the OOD doesn't exist, because the missing-label problem more emphasize on the (non-random) *sampling* or *labeling* process, e.g., user will not randomly rate his/her interacted items, but selectively rate some).
> > >
> > > - In summary, the reviewer mentioned OOD problem is definitely practical and important, but is orthogonal to our studied SSL with MNAR labels problem. To avoid misunderstanding that we use a strong assumption, we feel the necessity to change $P_{train}$ to $P_{ideal}$ (please kindly find more details below). We consider the combination of OOD and selection bias to be an interesting research direction for the future work.
> > >
> > > Next, we would like to briefly discuss, **What makes our unique contribution?**
> > >
> > > - To our best knowledge, all previous work studied selection bias in recommender system are limited to the fixed user-item pairs.
> > >   - That is, given a finite user-item dataset $D_{ideal}=\\{u_1, u_2, \ldots, u_m\\} \times \\{i_1, i_2, \ldots, i_n\\}$ as the ideal training set, due to the MNAR labels, we can only observe the ratings from a subset of user-item pairs, i.e., $D_{obs}\subset D_{ideal}$.
> > >   - In this paper, we find that **previous widely-used inverse propensity scoring (IPS) and doubly robust (DR) methods can only achieve unbiasedness when $D_{test}=D_{ideal}$, not even under $P_{test}=P_{ideal}$**, that is, **cannot generalize well even for a new user $u_{m+1}$ or a item $i_{n+1}$ that sampling from the same distribution of $P_{ideal}$.**
> > >   - Technically, this is because previous work only consider the randomness of $O_{u, i}$, not for the rest variables $X_{u, i}, Y_{u, i}$. By treating all variables are random, we find an interesting theoretical gap between previous finite-population DR and our super-population DR (see Lemma 4.1 v.s. Theorem 4.3, and Corollary 4.2 v.s. Corollary 4.4), as the core contribution of our paper.
> > >
> > > > Second, the rebuttal contains many elusive concepts. For example, what is 'target data distribution in training distribution'?
> > >
> > > Thank you for raising this concern. We acknowledge that this is vague, but what we intend to say is that we only need to assume that the ideal distribution (note that the training distribution with MNAR labels is sampled from this ideal distribution) and the test distribution are the same, i.e., $P_{test}=P_{ideal}$, without requiring the datasets to be exact the same under finite population, i.e, $D_{test}=D_{ideal}$. In short, we **extend the same _dataset_ requirement to same _distribution_ requirement,** and we consider further incorporating OOD to allow $P_{test}\neq P_{ideal}$ is natural, by combining the existing methods for overcoming distribution shifts.
> > >
> > > > What is 'popular user'? I assume the authors use this term to mean 'active user'.
> > >
> > > Thank you for pointing out this issue! You are absolutely right that it should be 'active user' rather than 'popular user'.

---

> > > > ### Author Response · Authors · 2025-08-07
> > > >
> > > > > Besides, the results on the first set (popular users & items) of KuaiRec show DR-JL outperforms the proposed method w.r.t. Recall and NDCG.
> > > >
> > > > Thank you so much for your careful reading. We checked the experimental log and found that this is actually true result. We are honest to say testing on the first set (popular users & items), by tuning both DR-JL and our method entirely, their performance are comparable. Please also kindly note that our method outperform the DR-JL in almost the remaining test scenarios.
> > > >
> > > > > In addition, it is elusive why testing on popular and unpopular items separately can demonstrate that the proposed method can handle nonstationarity between training and test sets.
> > > >
> > > > - Thank you for the question. This is a misunderstanding - we never *testing* on popular and unpopular items separately, but instead *training* on popular and unpopular items separately, which aims to demonstrate our method can perform well even in the violation of $P_{train}=P_{test}$.
> > > >
> > > > > Last but not least, what is the necessity of introducing a new baseline, DR-JL?
> > > >
> > > > - Thanks for the comment. DR-JL is exactly the 'DR' in our original manuscript - we are sorry for the ambiguity, so we not introduce new baseline. More specifically, DR-JL is the most popular baseline developed upon the doubly robust (DR) estimator, but as discussed before, DR-JL limits to the finite population. Our paper extend the widely used DR-JL to the super-population, named Super-DR. That's why we compared DR-JL here to make a fair comparison.
> > > >
> > > > > For W2, my concern is that the errors made by the propensity model may degrade the final accuracy. However, the authors compare two types of methods, both of which involve a propensity model, and the results show that these two types of methods achieve comparable accuracy. I don't understand how such results can address my concern.
> > > >
> > > > - Thank you for the great question! During our rebuttal, we intend to say - under varying propensity training paradigms, our method stably outperform DR-JL, showing that our performance gain is insensitive to the specific propensity training method. To fully address your concern, **we add new experiments with clearly printed propensity learning accuracy (marked as AUC$_p$, Precision$_p$, Recall$_p$, F1$_p$) and final accuracy of the prediction model (marked as AUC, NDCG@5, Recall@5),** in which Ours (1-layer), Ours (2-layer), Ours (3-layer) indicates the propensity model in our method is trained via 1-layer, 2-layer, and 3-layer NN, respectively.
> > > >
> > > > **Coat**
> > > > |Num of layers|AUC$_p$|Precision$_p$|Recall$_p$|F1$_p$|AUC|NDCG@5|Recall@5|
> > > > |-|-|-|-|-|-|-|-|
> > > > |Ours (1-layer)|0.704|0.129|0.667|0.216|0.753|0.661|**0.649**|
> > > > |Ours (2-layer)|**0.711**|**0.172**|0.594|**0.267**|**0.757**|**0.667**|0.637|
> > > > |Ours (3-layer)|0.690|0.114|**0.728**|0.196|0.740|0.655|0.624|
> > > > |Best baseline|0.709|0.153|0.582|0.242|0.750|0.654|0.630|
> > > >
> > > > **Yahoo**
> > > > |Num of layers|AUC$_p$|Precision$_p$|Recall$_p$|F1$_p$|AUC|NDCG@5|Recall@5|
> > > > |-|-|-|-|-|-|-|-|
> > > > |Layer1_JL|0.758|0.133|**0.402**|0.200|0.674|0.654|0.450|
> > > > |Layer2_JL|0.752|0.181|0.316|0.230|**0.686**|**0.667**|0.463|
> > > > |Layer3_JL|0.757|**0.222**|0.263|**0.241**|0.684|0.665|**0.464**|
> > > > |Best baseline|**0.764**|0.187|0.318|0.236|0.679|0.662|0.459|
> > > >
> > > > **KuaiRec**
> > > > |Num of layers|AUC$_p$|Precision$_p$|Recall$_p$|F1$_p$|AUC|NDCG@50|Recall@50|
> > > > |-|-|-|-|-|-|-|-|
> > > > |Layer1_JL|0.632|0.049|**0.879**|0.093|0.826|0.631|0.667|
> > > > |Layer2_JL|0.647|0.061|0.671|0.112|0.828|**0.640**|**0.680**|
> > > > |Layer3_JL|**0.653**|0.068|0.543|0.121|**0.829**|0.636|0.675|
> > > > |Best baseline|0.648|**0.080**|0.388|**0.133**|0.824|0.632|0.671|
> > > >
> > > > Note: the subscript $p$ means that this metric evaluates the propensity model.
> > > >
> > > > From above, we find **our methods stably outperform the best baseline, even with a larger error on the trained propensity model.**
> > > >
> > > > ***
> > > >
> > > > We are eager to hear your feedback. We’d deeply appreciate it if you could let us know whether your concerns have been addressed.

---

> > > ### Author Response · Authors · 2025-08-08
> > >
> > > Dear Reviewer gNuC,
> > >
> > > As the discussion deadline approaches, we are wondering whether our responses have properly addressed your concerns? Your feedback would be extremely helpful to us. If you have further comments or questions, we hope for the opportunity to respond to them.
> > >
> > > Many thanks,
> > >
> > > 19687 Authors

---

> > > > ### Author Response · Authors · 2025-08-09
> > > >
> > > > Dear reviewer gNuC,
> > > >
> > > > Since the discussion period will end in a few hours, we will be online waiting for your feedback on our rebuttal, which we believe has fully addressed your concerns.
> > > >
> > > > We would highly appreciate it if you could take into account our response when updating the rating and having discussions with AC and other reviewers.
> > > >
> > > > Thank you so much for your time and efforts. Sorry for our repetitive messages, but we're eager to ensure everything is addressed.
> > > >
> > > > Authors of # 19687

---

### Official Review · Reviewer_g41m · 2025-07-01

**Clarity:** 3
**Significance:** 4
**Originality:** 3
**Rating:** 5
**Confidence:** 3

**Summary:**

This paper introduces SuperDR, a novel doubly robust estimator for debiasing recommendation models under a more realistic super-population scenario, where the joint distribution of features and ratings remains the same across training and test sets, but the specific user-item pairs may not overlap. This setting contrasts with the conventional exact matching assumption commonly made in prior work. The key insight is that, under the super-population setting, existing DR estimators remain biased due to an uncontrolled covariance term, even when the imputation and propensity models are correctly specified. To address this, the authors propose a correction mechanism through a learnable parameter $\epsilon$, which is added to the imputation model. This adjustment aims to reduce the empirical covariance $\hat{\text{Cov}}\Big({\hat{p}_{u, i} - o_{u, i} \over \hat{p}_{u, i}}, e_{u, i} - \hat{e}_{u, i}\Big)$. The paper presents supporting theoretical results—including a generalization error bound—and demonstrates, through experiments on three datasets (Coat, Yahoo! R3, and KuaiRec), that SuperDR outperforms existing debiasing methods.

**Questions:**

Nothing

**Ethical Concerns:**

["NO or VERY MINOR ethics concerns only"]

**Final Justification:**

The paper is well-organized, and the proposed method shows strong performance compared to the baselines. The authors also responded sincerely and appropriately during the discussion period to the weaknesses and related questions raised by other reviewers. In particular, I believe the authors provided a sufficiently convincing response to the concern raised by another reviewer regarding the Assumption of Equal Joint Distributions. However, the concern about Dependence on Propensity Pre-training still appears to remain partially unaddressed. That said, the motivation behind the problem, the proposed solution, and the logical progression presented by the authors offer a meaningful contribution to the field.

**Limitations:**

The authors properly address the limitations in Section 6.

**Paper Formatting Concerns:**

There is no major formatting issue.

**Quality:**

4

**Strengths And Weaknesses:**

- **Strengths**
    - The shift from exact matching to super-population is realistic and meaningful, especially for large-scale industrial recommendation systems, where user-item overlap between training and test sets is uncommon.
    - The paper establishes a strong theoretical foundation, with clearly articulated lemmas, corollaries, and proofs. Notably, the identification and treatment of the covariance term addresses an important and underexplored aspect of doubly robust (DR) estimation.
    - The correction to the imputation model using $\epsilon (o_{u, i} - \hat{p}_{u, i})$ is elegant and well-motivated by targeted maximum likelihood estimation (TMLE).
    - The derivation of the generalization error bound incorporating the new covariance term and empirical Rademacher complexity is both novel and informative.
    - Experiments on multiple datasets consistently demonstrate performance improvements over a broad set of DR- and IPS-based baselines.
    - The sensitivity study on the learning rate for $\epsilon$ and sample ratio shows the method’s robustness under different bias severities and hyperparameter configurations.

- **Weaknesses**

    Nothing

---

> ### Author Rebuttal · Authors · 2025-07-31
>
> Response: Thank you for your time and dedication in reviewing our manuscript! We are really glad that you are enjoying our paper and give our paper very positive comments! We believe that our study makes an important contribution to the field of debiasing in RecSys and causal inference, and we hope that you will support our research in subsequent discussions.

---

### Official Review · Reviewer_UUJ9 · 2025-07-03

**Clarity:** 3
**Significance:** 3
**Originality:** 3
**Rating:** 5
**Confidence:** 3

**Summary:**

The paper proposes a novel doubly robust estimator with a joint learning algorithm to address the limitation of selection bias in designing the recommender systems.

**Questions:**

Can this method be applied in a real-time setting where data arrives sequentially? In other words, is the proposed approach capable of efficiently adapting or updating its predictions as new data becomes available, without requiring full retraining?

**Ethical Concerns:**

["NO or VERY MINOR ethics concerns only"]

**Limitations:**

See above.

**Quality:**

3

**Strengths And Weaknesses:**

The paper is well-organized and easy to follow. It presents strong results on large-scale datasets, which highlight the effectiveness and practical relevance of the proposed method. Additionally, the inclusion of a generalization bound provides valuable theoretical support, offering insights into the method's expected performance beyond the training data. However, further clarification or empirical validation of the bound would strengthen the contribution even more.

---

> ### Author Rebuttal · Authors · 2025-07-31
>
> We thank the reviewer for efforts in evaluating our work and helpful suggestions. Below, we will try our best to address your concerns and question.
>
> > **W1:** Further clarification or empirical validation of the bound would strengthen the contribution even more.
>
> **Response to W1:** We thank the reviewer for the useful suggestions.
>
> - Further clarification of error bounds:
> Given the propensity model $\hat p_{u, i}$ and imputation model $\hat e_{u, i}$, satisfying $\hat p_{u, i}\geq K_{\psi}$ and $|e_{u, i}-\hat e_{u, i}|\leq K_{\phi}$, then with probability at least $1-\eta$, we have
>
> $$L_{\text{ideal}}(\theta) \leq \underbrace{{\frac{1}{|\mathcal{D}|} \sum_{(u, i) \in \mathcal{D}}\big[\hat e_{u, i}+\frac{o_{u, i}(e_{u, i}-\hat e_{u, i})}{\hat{p}_{u, i}}\big]}} _{\text{empirical DR loss}} $$
>
> $$ +  ~\underbrace{\frac{1}{|\mathcal{D}|} \sum_{(u, i) \in \mathcal{D}}\left|1-\mathbb{E}\left[\frac{o_{u, i}}{\hat p_{u, i}} | x_{u, i}\right]\right| \cdot\left|\mathbb{E}\left[e_{u, i} \mid x_{u, i}\right]-\mathbb{E}\left[\hat e_{u, i} \mid x_{u, i}\right]\right|} _{\text{empirical risks of imputation and propensity models}}$$
>
> $$ + ~\underbrace{\left|\frac{1}{|\mathcal{D}|} \sum_{(u, i) \in \mathcal{D}} \operatorname{Cov}\left(\frac{o_{u, i}-\hat p_{u, i}}{\hat p_{u, i}}, e_{u, i}-\hat e_{u, i}\right)\right|} _{\text{the empirical covariance}}$$
>
> $$ + \underbrace{\left(1+\frac{1}{K_\psi}\right)\left(2 \mathcal{R}(\mathcal{F})+K_\phi \sqrt{\frac{18 \log (4 / \eta)}{|\mathcal{D}|}}\right)} _{\text{tail bound controlled by empirical Rademacher complexity and sample size}},$$
>
> which includes four terms: the empirical DR loss, the empirical risks of imputation and propensity models, the empirical covariance, and the tail bound. Even if the $\hat p_{u,i} = p_{u,i}$ or $\hat e_{u,i} = e_{u,i}$, the empirical covariance does not vanish in the super-population scenario. **The previous DR-based method can only control the first two terms, while our method can further control the empirical covariance by subtly modifying the imputation model (introducing one learnable parameter), without adding new modules and increasing time complexity.**
>
> - Further empirical validation of error bounds:
>   - In our manuscript, we have demonstrated that **our method can control empirical covariance on the observed samples on three widely used real-world datasets.**
>   - Furthermore, we validate **empirical covariance control on the whole dataset by conducting a semi-synthetic experiment on the Movielens 100k dataset.** Following previous studies [1-4], we use matrix factorization to generate the full binary rating matrix as the ground truth, the propensities for generating MNAR observed data, as well as the embedding for each user and item. Then we randomly select 50% / 70% users and items into our training data and train DR-JL, DCE-DR, and our method on the training data, finally calculating the empirical covariance (EC) on the whole dataset.
>
> The results are shown below:
> ||Ratio = 50%||Ratio = 70%||
> |-|-|-|-|-|
> |Method|AUC|EC on Whole Data|AUC|EC on Whole Data|
> |DR-JL|0.8855|-3.3429|0.9104|-3.0932|
> |DCE-DR|0.8941|-1.9507|0.9169|-2.0315|
> |Ours|**0.9017**|**-0.9428**|**0.9214**|**-1.5318**|
>
> **The results show that our method can control EC the most, which verifies the validity of the Controllability of Empirical Covariance (Thm 4.7), as well as we may a tighter error bounds.**
>
> > **Q1:** Can this method be applied in a real-time setting where data arrives sequentially? In other words, is the proposed approach capable of efficiently adapting or updating its predictions as new data becomes available, without requiring full retraining?
>
> **Response to Q1:** We thank the reviewer for the insightful question. This is a very common scenario in practice. For example, in the CTCVR scenario, the new MNAR data comes daily. **To evaluate whether our method can be efficiently applied in this scenario, we propose two ways to efficiently update our predictions:**
> - We store the parameters in models and optimizers at the end of training, and fine-tune them when new data comes (named "continue training").
> - We freeze the parameters in prediction, imputation, and propensity model, and add a learnable shallow residual structure, which is zero-initialized when new data comes (named "adding residual").
> - Also, we consider the full retraining of all parameters on both old and new data as the oracle baselines.
>
> **We add experiments on the industrial dataset KuaiRec.** To process the data, we first randomly split 80% users and items to construct the training set to make sure we do not see all users and items during training. Then we consider the following two scenarios:
> - **The new data has the same distribution compared to the training data (Consistent in distribution).** We randomly split the user set and item set into three parts with a ratio of 50/25/25. First, we train our method on the data with 50% users and 50% items, and then retrain the model on the remaining data sequentially, i.e., twice in total.
>
> - **The new data has a different distribution compared to the training data (Inconsistent in distribution).** We randomly split the user set and item set into two parts with a ratio of 50/50, and train our method on the first dataset. Then we assign a propensity according to the item frequency and the user's active level to sample the first half of users and items into the first time retraining, and the remaining into the second time retraining. We control the sampling process to ensure the sample number is the same in the two retrainings.
>
> For all methods, we test on the MAR dataset. The results on the **KuaiRec** dataset are shown below. We bold the better results between ours and DR-JL.
>
> |Data||Consistent in distribution|||Inconsistent in distribution|||
> |-|-|-|-|-|-|-|-|
> |||AUC|NDCG@50|Recall@50|AUC|NDCG@50|Recall@50|
> |Train Set1 (50%)|Ours|**0.825**|**0.636**|**0.678**|**0.825**|**0.636**|**0.678**|
> ||DR-JL|0.817|0.623|0.663|0.817|0.623|0.663|
> |Train Set2 (25%)|Continue training on ours|**0.819**|**0.628**|**0.664**|**0.821**|**0.633**|**0.669**|
> ||Continue training on DR-JL|0.808|0.617|0.647|0.811|0.631|0.666|
> ||Adding residual on ours|**0.811**|**0.614**|**0.648**|**0.821**|**0.640**|**0.680**|
> ||Adding residual on DR-JL|0.805|0.599|0.639|0.816|0.620|0.653|
> ||Full training on ours|**0.826**|**0.638**|**0.679**|**0.827**|**0.640**|**0.679**|
> ||Full training on DR-JL|0.819|0.625|0.665|0.821|0.626|0.665|
> |Train Set3 (25%)|Continue training on ours|**0.819**|**0.629**|**0.660**|**0.729**|**0.492**|**0.504**|
> ||Continue training on DR-JL|0.817|0.617|0.654|0.712|0.457|0.449|
> ||Adding residual on ours|**0.824**|**0.627**|**0.658**|**0.725**|**0.479**|**0.506**|
> ||Adding residual on DR-JL|0.823|0.614|0.653|0.715|0.450|0.449|
> ||Full training on ours|**0.828**|**0.640**|**0.679**|**0.828**|**0.640**|**0.679**|
> ||Full training on DR-JL|0.822|0.627|0.668|0.822|0.627|0.668|
> |
>
> We find that our methods can efficiently adapt our predictions as new data becomes available. **An interesting finding is that when there is a large gap between the new data and the original training data (such as cold users and items, as shown in the "Train Set3" in the "Inconsistent in distribution" scenario), though the performance of our method will drop, our method significantly outperforms DR-JL. One possible reason is that our method deals with a super-population scenario, which is more robust toward a huge distribution shift.**
>
> We sincerely thank you for your feedback, and welcome any further technical advice or questions on this work. We will do our best to address your concerns.
>
> ---
> References
>
> [1] Recommendations as Treatments: Debiasing Learning and Evaluation. ICML 2016.
>
> [2] Doubly Robust Joint Learning for Recommendation on Data Missing Not at Random. ICML 2019.
>
> [3] Enhanced Doubly Robust Learning for Debiasing Post-Click Conversion Rate Estimation. SIGIR 2021.
>
> [4] Be Aware of the Neighborhood Effect: Modeling Selection Bias under Interference for Recommendation. ICLR 2024.

---

### Note · Authors · 2025-08-13

Dear reviewers and AC,

We sincerely thank all reviewers and AC for their great effort and constructive comments on our manuscript. We are encouraged that **3 out of 4 reviewers are currently on the positive side, with a rating 5 (conf 3), 5 (conf 3), and 4 (conf 3)**, recognizing our paper
- “Valuable theoretical support, strong results on large-scale datasets” (Reviewer UUJ9),
- “Realistic and meaningful” problem with “strong theoretical foundation” and “elegant” proposed method (Reviewer g41m),
- “Fills an important gap and better reflects real-world applications, theoretical results are elegant and insightful” (Reviewer FSqV).
- **Notably, Reviewer g41m gives our paper very positive comments with no weaknesses mentioned.**

Meanwhile, we noticed that Reviewer gNuC gave an initial rating 3 (conf 2) with no final rating and justification yet. After acknowledging on 4 Aug, Reviewer gNuC's current concerns are mainly on (1) Assumption of Equal Joint Distributions, and (2) Dependence on Propensity Pre-training.

> The covariate drift or nonstationarity of data distribution means the preference shift of users or items becoming outdated.
- **We were wondering if there was a misreading – the mentioned covariate drift problem is practical, but is orthogonal to our studied problem, i.e., semi-supervised learning (SSL) with missing-not-at-random (MNAR) labels.**
- In SSL papers, people are more concerned with covariate drift $P_{obs}(X, Y)=P(X, Y \mid O=1) \neq P(X, Y)$ due to MNAR labeled data, where $O$ is the labeled data indicator. They usually assume that the covariate drift doesn't exist, because the missing-label problem places more emphasis on the (non-random) labeling process, e.g., a user will not randomly rate his/her interacted items. But these methods (including ours) can be naturally extend to the covariate drift scenarios.

> The errors made by the propensity model may degrade the final accuracy.
- We add experiments under varying propensity training paradigms, demonstrating that **our performance gain is insensitive to the propensity training method and model structure.**

We are confident that our responses can thoroughly address Reviewer gNuC's concerns. We kindly remind that Reviewer gNuC has not replied to our further comments, which were uploaded 59 hours before deadline (9 Aug). Thus, we respectfully ask AC and reviewers to consider this when making recommendations. Thank you for your invaluable time and effort.

Many thanks,

The Authors

---

### Decision · Program_Chairs · 2025-09-17

**Decision:**

Accept (poster)

**Comment:**

This paper considers the problem of selection bias. They go beyond the setting of exact matching and instead assume only that the joint distribution of the feature and the rating is the same in the training and test set. They show that in this broader setting, the previous DR estimator is biased due to an additional covariance term. They propose SuperDR which adds a learnable “correction” parameter. They provide sound and nontrivial theoretical guarantees and demonstrate improved performance on a variety of datasets. This paper is novel, well-written, and a nice contribution.